# Early commitment and robust differentiation in colonic crypts

Beáta Tóth[1,†], Shani Ben-Moshe[1,†], Avishai Gavish[2,†], Naama Barkai[2] & Shalev Itzkovitz[1,*] ID

## Abstract

Tissue stem cells produce a constant flux of differentiated cells with distinct proportions. Here, we show that stem cells in colonic crypts differentiate early to form precisely 1:3 ratio of secretory to absorptive cells. This precision is surprising, as there are only eight stem cells making irreversible fate decisions, and so large stochastic effects of this small pool should have yielded much larger noise in cell proportions. We use single molecule FISH, lineage-tracing mice and simulations to identify the homeostatic mechanisms facilitating robust proportions. We find that Delta-Notch lateral inhibition operates in a restricted spatial zone to reduce initial noise in cell proportions. Increased dwell time and dispersive migration of secretory cells further averages additional variability added during progenitor divisions and breaks up continuous patches of same-fate cells. These noise-reducing mechanisms resolve the trade-off between early commitment and robust differentiation and ensure spatially uniform spread of secretory cells. Our findings may apply to other cases where small progenitor pools expand to give rise to precise tissue cell proportions.

**Keywords** Delta-Notch; design principles; noise reduction; robustness stem cells

**Subject Categories** Development & Differentiation; Quantitative Biology & Dynamical Systems; Stem Cells

**Mol Syst Biol. (2017) 13: 902**

## Introduction

Tissue stem cells maintain their numbers while continuously producing distinct types of differentiated cells (Morrison *et al*, 1997; Crosnier *et al*, 2006; van der Flier & Clevers, 2009; Clevers, 2013). These cell types often need to be produced at optimally tuned proportions. The homeostatic control of differentiated cell production is challenging, as stem cells operate in a regime of exponential proliferation, rendering cell numbers highly sensitive to biological noise (Lander *et al*, 2009; Hart *et al*, 2012).

The mouse intestine is a classic model system for studying tissue stem cell processes. The epithelium of both the small intestine and the large intestine (colon) consists of deep invaginations called crypts. At the bottom of the crypts, a small number of ~6–8 Lgr5[+] stem cells (SCs; Lopez-Garcia *et al*, 2010; Snippert *et al*, 2010; Kozar *et al*, 2013; Vermeulen *et al*, 2013; Ritsma *et al*, 2014) constantly divide to yield SCs and progenitors, termed transit amplifying (TA) cells. TA cells divide as they migrate upwards, yielding two distinct lineages, absorptive cells (enterocytes) and secretory cells. The post-mitotic differentiated cells continue to migrate and are eventually shed into the lumen. In the colon, secretory cells predominantly consist of goblet cells that secrete mucus, a key protective barrier against pathogens and mechanical stress (Atuma *et al*, 2001; Deplancke & Gaskins, 2001; Kim & Ho, 2010).

Despite some similarities, the small intestine and the colon differ in several important aspects. In the small intestine, differentiated cells mostly operate outside of crypts in large finger-like protrusions called villi, which are each fueled by 6–10 surrounding crypts. In contrast, the colon lacks villi and the differentiated cells operate within the crypt. The microenvironments of these two tissues are also dramatically different. While the small intestine is richer in nutrients, the distal colon is the most abundant site of microbiota in our body and contains orders of magnitude higher numbers of bacteria (Bäckhed *et al*, 2005; Sender *et al*, 2016). To protect against this hostile microenvironment, the proportions of goblet cells are much higher in the colon (25% goblet cells versus 8% in the small intestine), producing a sevenfold thicker protective mucous layer (Atuma *et al*, 2001).

Since the entire colonic crypt, including the stem cell compartment, is exposed to the potentially toxic microbiota-rich lumen, it seems that production of precise number of goblet cells, that would be functional throughout the crypt axis, is an important design principle in the colon. Low goblet cell numbers may result in inhomogeneous mucosal coverage, potentially leading to epithelial damage, as is indeed observed in ulcerative colitis (Gersemann *et al*, 2009; Zheng *et al*, 2011). Excessive goblet cell production may also be disadvantageous, as it would come at the expense of enterocytes, compromising absorbing capacities (Lam *et al*, 2002). The precision in generating fixed proportions of goblet cells and enterocytes is intimately related to the stage along the SC-TA hierarchy at which commitment to one of these two fates occurs.

1 Department of Molecular Cell Biology, Weizmann Institute of Science, Rehovot, Israel
2 Department of Molecular Genetics, Weizmann Institute of Science, Rehovot, Israel
*Corresponding author. Tel: +972 9343104; E-mail: shalev.itzkovitz@weizmann.ac.il
†These authors contributed equally to this work

When considering cell fate commitment, one can envision two opposing strategies. In the first strategy, SCs generate progenitors that first divide to amplify their numbers and differentiate only upon cell cycle exit. Under such "'late commitment", there is a clear division in time and space between proliferation and differentiation (Fig 1A, left). Under this strategy, fate choice is made by a large progenitor pool, thus minimizing stochastic effects that could lead to variability in the proportions of cell types. An opposite strategy entails fate commitment at the very exit from the SC compartment, before TA clonal expansion. Under such "early commitment", differentiation could partially or even fully overlap proliferation (Fig 1A, right). Early commitment can facilitate ample time for cell maturation and can also enable production of functional goblet cells at lower crypt positions. However, as we will demonstrate, decisions in an early commitment scenario are made by a small number of cells, exposing the system to stochastic fluctuations that could generate profound variability in the proportions of differentiated cells among crypts.

Previous studies suggested that early commitment may be the predominant mode of differentiation in the small intestinal crypts (Cheng, 1974; Cheng & Leblond, 1974; Bjerknes & Cheng, 1981a, 1999). Bjerknes and Cheng traced the differentiation state of single-cell-derived clones induced by chemical mutagenesis in the small intestine (Bjerknes & Cheng, 1999) and found very few clones that contained more than one cell type. This was a clear indication that commitment precedes clonal expansion in this tissue. Goblet cells are clearly visible at lower crypt positions in the colonic crypts and have also been shown to incorporate 3H-thymidine (Chang & Leblond, 1971a), suggesting that fate commitment may be an early event in this tissue as well; however, clonal tracing to directly address commitment stage has not been performed in the colon. Such experiments are important, as goblet cells could also be generated at higher crypt positions, subsequently migrating toward the crypt base, as has been shown for the small intestinal Paneth cells (Bjerknes & Cheng, 1981b).

Here, we explore the trade-off between early commitment and robust differentiation in the mouse colonic crypt. We first trace the fate of rare clones using Lgr5-Cre/Confetti, as well as CAG-Cre/Confetti mice and show that, as in the small intestine, commitment in the colonic crypt is restricted to the very exit from the stem cell compartment, before TA clonal expansion has commenced. We describe a fundamental vulnerability of such early commitment entailing sensitivity to biological noise that is associated with the small numbers of progenitors making fate decisions. This sensitivity can give rise to large fluctuations in cell proportions among crypts, potentially generating epithelial patches depleted of mucosal protection. We demonstrate that efficient homeostatic mechanisms, involving Delta-Notch lateral inhibition that is restricted to a confined commitment zone, as well as dispersive goblet cell migration, buffer this variability. These mechanisms ensure early commitment yet robust differentiation.

# Results

## A fundamental trade-off between robustness and timely production of differentiated cells

To characterize the impact of commitment stage on the differentiation process, we simulated a stereotypical SC-maintained tissue inspired by the colonic crypt. The system is composed of a cylindrical hexagonal lattice with three compartments—a SC compartment comprised of a single row of 8 SCs, intermingled with 8 non-dividing niche cells representing deep secretory cells (Rothenberg *et al*, 2012), six rows of proliferating progenitors (the TA compartment), and 15 rows of post-mitotic cells (the differentiated compartment, Fig 1B and C). In our model, SCs and progenitors all divide with equal rates of once every 2.5 days and cell movement is generated by mitotic pressure—whenever a cell divides, the entire column above it is pushed upwards (Appendix). In our simulations, SCs can replace both adjacent SCs and TA cells, according to the neutral drift dynamics demonstrated for crypt SCs (Lopez-Garcia *et al*, 2010; Snippert *et al*, 2010; Klein & Simons, 2011; Ritsma *et al*, 2014).

A key feature of the system is that cells accelerate as they are displaced upwards, since cells at higher crypt positions have more dividing cells below them (Fig 1D). As a result, cells spend less than half of their post-SC lifetime in the differentiated compartment, even though it is considerably larger than the TA compartment. If cells require time to build up the differentiated machinery needed for proper function, late commitment commencing at the differentiated compartment could result in the extrusion of cells from the tissue before reaching a fully functional state. In contrast, early commitment allows ample time for cells to mature by the time they reach the upper crypt positions and to function in the lower crypt positions as well.

The accelerated cell movement has an opposite effect on variability in cell proportions. By the time eight cells feed the bottom row of the TA compartment with new cells, 80 cells feed the differentiated compartment (Appendix). Stochastic early commitment of eight progenitors to a secretory fate (with probability 25%, the fraction observed in the distal colon) would yield a binomially distributed $2 \pm 1.22$ secretory clones per typical SC division time. These clones will be amplified 10-fold to yield $20 \pm 12$ secretory cells feeding the differentiated compartment, with a large coefficient of variation (CV) of ~0.6. Under late commitment, 80 rather than eight cells make the stochastic choice during the same time period, yielding $20 \pm 3.8$ secretory cells, with a considerably smaller CV of 0.19. Late commitment should thus yield significantly lower variability than early commitment.

To quantify the effect of commitment stage on variability, we simulated early and late commitment by stochastically drawing secretory or absorptive fates either as cells enter the TA compartment (Fig 1C), or as they enter the bottom row of the differentiated compartment (Fig 1B). We sampled simulated epithelial patches of 20–100 cells and calculated the variability in secretory cell proportions. Indeed, we found that early commitment yielded profoundly higher variability in goblet cell proportions, compared to late commitment (Fig 1B–E, simulated CV = 0.57 for early commitment versus 0.24 for late commitment, Appendix). Thus, early commitment enables sufficient time for cell maturation but is less robust, as it generates considerable variability in the proportion of secretory cells among crypts (Fig 1F).

## Commitment in the colon is early, preceding clonal expansion

Determining at which stage commitment occurs in colonic crypts requires lineage tracing of the progenies of the first TA progenitor, at the very exit from the SC compartment. By irreversibly labeling

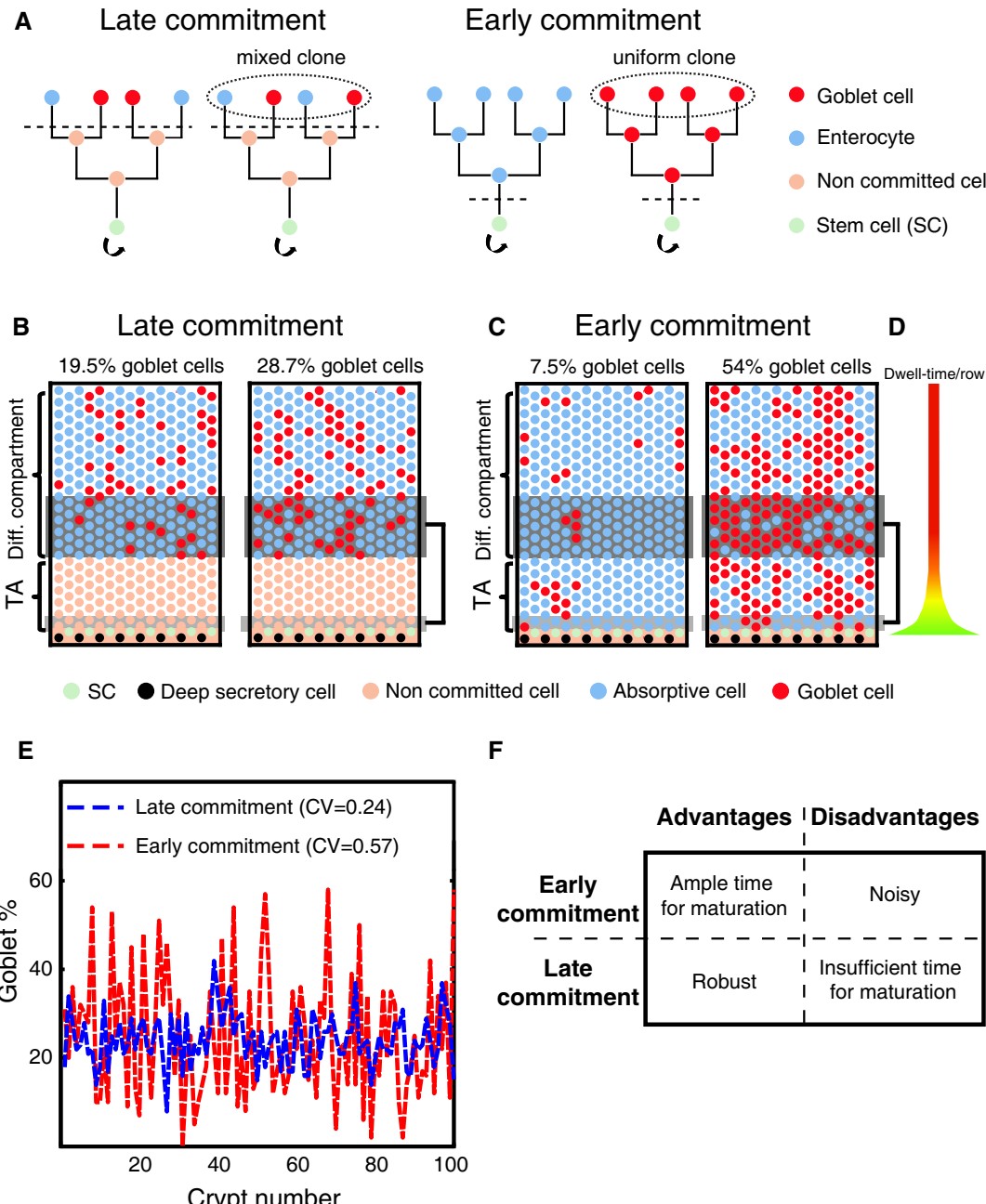

**Figure 1. Commitment stage affects variability in cell proportions.**

A   Late commitment entails clonal expansion followed by commitment and differentiation (left). Early commitment entails commitment preceding clonal expansion (right). Early, but not late commitment yields single-cell clones that are uniform in their differentiation fate.

B   Two simulated crypts demonstrating low variability in cell proportions when commitment is late. Crypt topology is represented by a monolayer of epithelial cells covering a cylindrical crypt (the left and right columns in sheet are in contact). Dark gray boxes highlight the 80 cells that enter the differentiated compartment over a period in which 8 SC progenies feed the bottom TA row (light gray boxes).

C   Two simulated crypts demonstrating high variability in cell proportions when commitment is early. Dark gray boxes highlight the 80 cells that enter the differentiated compartment over a period in which 8 SC progenies feed the bottom TA row (light gray boxes).

D   Cells spend more time in the lower crypt rows. Color and width of the graded bar correspond to the dwell time of cells in each row in the crypt simulations presented in (B, C) (green, long dwell time at the bottom of the TA compartment; red, short dwell time in the differentiated compartment).

E   The simulated coefficient of variation of goblet cell proportions among crypts is much smaller under late commitment compared to early commitment. Dashed red line, distribution of goblet cell proportions based on a stochastic early commitment model (see Appendix). Dashed blue line, distribution of goblet cell proportions based on a stochastic late commitment model.

F   Advantages and disadvantages of early versus late commitment. Early commitment allows ample time for cell maturation but yields significantly higher variability in cell proportions among crypts.

such early progenitors and tracing their progenies, one can deduce early versus late commitment from the differentiation status of the clonal cells—late commitment is expected to yield abundant mixed clones, containing both goblet cells and enterocytes. In contrast, early commitment should give rise to predominantly uniform clones (Fig 1A). While several genes have been shown to be expressed at the TA compartment (van Es *et al*, 2012), no gene has yet been shown to be exclusively expressed at the bottom TA layer, thus prohibiting such labeling scheme. To overcome this hurdle, we sought to use two alternative lineage-tracing mouse models—Lgr5-CreERT2/Confetti and CAG-Cre/Confetti mice.

Lgr5 is exclusively expressed in the SC compartment (Muñoz *et al*, 2012). Lgr5 SCs either divide symmetrically to generate two SCs, or are extruded due to symmetric divisions of other SCs. Clones that include progenies that have divided within the SC compartment are less informative to the question of commitment stage, since they can yield mixed clones by the mere fact that they emanate from more than a single extruded SC. Indeed, we calculated that the majority of clones emanating from single $Lgr5^+$ labeled cells would have undergone at least one division within the SC compartment (Appendix). In contrast, clones in which all cells were outside the SC compartment are expected to have a 50% chance of emanating from a single labeled SC that was extruded before it had a chance to divide within the SC compartment (Fig 2A and B). These clones would thus be informative to the question of commitment stage.

We traced the fate of clones emanating from individual colonic $Lgr5^+$ stem cells using Lgr5-CreERT2/Confetti mice (Snippert *et al*, 2010). Upon tamoxifen injection, rare $Lgr5^+$ cells became irreversibly labeled. We extracted intestinal tissue from mice sacrificed at sequential time points (40 h, 2, 3, 4, 5, 7, 10, and 14 days postinduction) and observed clones of increasing sizes and positions along the vertical crypt axis (Fig 2A and B). Clone size dynamics within the stem cell compartment followed the neutral drift pattern previously documented, according to which individual stem cells either divide within the SC zone or are extruded due to mitotic divisions of other SCs (Appendix Fig S1, Table EV1). Analysis of the clonal growth dynamics revealed an average division rate once every 2.6 days (0.27/day, 95% C.I. 0.23–0.29/day, Appendix Fig S1B, Table EV1). To analyze the differentiated fate of clonal cells, we used single molecule fluorescence *in situ* hybridization (smFISH, Itzkovitz *et al*, 2011) to measure the expression of Gob5 and Fabp2, differentiation markers for goblet cells and enterocytes, respectively (Appendix Fig S2A). We found that more than 93% of $Gob5^-$ cells were Fabp2 high (Appendix Fig S2B), in line with previous work demonstrating that cell types other than goblet cells and enterocytes are extremely rare in the distal colon (Chang & Leblond, 1971b).

We found that most clones in the colonic crypts were uniformly either all $Gob5^+$ or $Gob5^-$ (70% uniform clones, 30% mixed clones, Fig 2C and D, Table EV2). This fraction was significantly larger than that expected under late commitment (32% uniform clones, $P < 10^{-16}$, Fig 2D, Materials and Methods). As expected, the fraction of mixed clones was larger when including clones with progenies within the SC compartment (55% mixed clones, Appendix Fig S3, Table EV2).

Since around 50% of the clones analyzed could still consist of progenies of SCs that have divided within the SC compartment, subsequently having both SC progenies extruded, the Lgr5-Confetti model may overestimate the fraction of mixed clones that emanate

from single extruded SCs. We therefore sought an additional mouse model that would enable tracing the progenies of single divisions throughout the crypt axis. To this end, we generated CAG-Cre/Confetti mice, where clones originated at any position along the crypt axis (Hayashi & McMahon, 2002; Lei & Spradling, 2013; Fig 2E and F) and traced them for 2 or 3 days. Indeed, the fraction of uniform clones was substantially higher (95%) in this mouse model. These results indicate that fate commitment in the colon occurs at the very exit from the SC compartment, predominantly before the first TA division occurs. Using smFISH, we detected extensive co-expression of proliferation and differentiation markers (EdU, Ki67, and Gob5, Fig 2G), as previously also observed by 3H-thymidine incorporation (Chang & Leblond, 1971a). This indicates that differentiation and proliferation overlap in the colonic crypt, as expected if commitment precedes clonal expansion.

## Cell proportions are significantly less variable than expected based on early commitment

We next asked whether early commitment indeed yields large variability in the differentiated cell proportions among crypts, as predicted by our stochastic commitment model simulations (Fig 1). To this end, we measured the proportion of goblet cells in multiple crypts. We found that variability in cell proportions among colonic crypts was threefold lower than that expected based on stochastic early commitment (Fig 3, Table EV3, CV = 0.21). In fact, the variability in goblet cell proportions was even lower than that expected from stochastic late commitment (Fig 1E, CV = 0.24). Thus, additional homeostatic mechanisms must operate in the colon to ensure minimal variability in goblet cell proportions among crypts.

## Delta-Notch lateral inhibition operates in the colon to reduce variability

A natural feedback mechanism for reducing variability is lateral inhibition. Under lateral inhibition, cells of the minor fate (secretory cells) inhibit their neighboring cells from becoming secretory cells (Fig 4A, Collier *et al*, 1996; Stamataki *et al*, 2011; Sancho *et al*, 2015). Theoretical studies by Lewis and colleagues demonstrated that lateral inhibition operating uniformly in all cells on a hexagonal lattice can naturally lead to a 3:1 ratio, regardless of SC output (Collier *et al*, 1996; Stamataki *et al*, 2011). Indeed, the Notch target transcription factor Hes1 and the Delta activator Math1 control the enterocyte-secretory fate decision (Yang *et al*, 2001; van Es *et al*, 2005, 2012; Shroyer *et al*, 2007; Fre *et al*, 2011; Pellegrinet *et al*, 2011; Noah & Shroyer, 2013; Kim *et al*, 2014). Notch-activated cells give rise to enterocytes, whereas Delta expressing cells become secretory cells.

The intestinal epithelium is unique in that it is not a static lattice. Rather, cell contacts are dynamically changing due to divisions and cell migrations, as cells constantly acquire new neighbors. Neighboring cells are also often clonally related (Fig 2B, C and E, Appendix Fig S4). Under lateral inhibition, one would expect to observe predominantly mixed clones (Fig 4B). Since we rarely observed such mixed clones (Fig 2C–F), and since Hes1 and Math1 can also operate cell autonomously without invoking lateral inhibition (Lewis, 2003; Monk, 2003; Kim & Shivdasani, 2011; Sancho *et al*, 2015), we next turned to examine whether colonic crypt cells

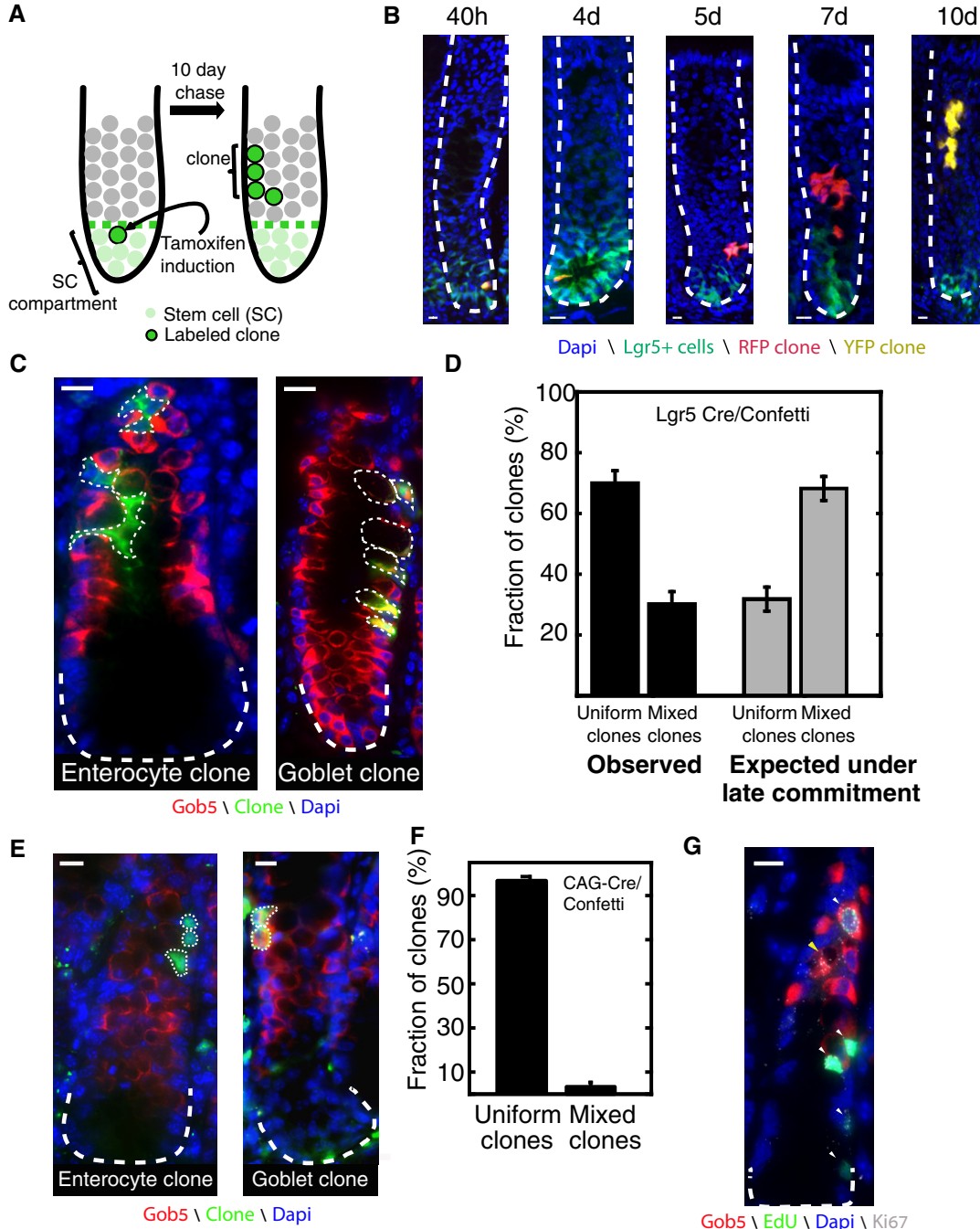

**Figure 2.  Commitment in colonic crypts is an early event.**

A  Single Lgr5[+] cells in Lgr5-CreERT2-Confetti mice are irreversibly labeled upon tamoxifen injection and the clones resulting from extruded SCs are traced over time. Green dashed line marks border of stem cell zone.

B  Representative clones at sequential time points post-induction. At short times after tamoxifen induction (40 h and 4 days), clones mostly consist of 1–2 Lgr5[+] SCs. At longer chase times (5, 7, and 10 days), clones with no progenies in the SC compartment progressively increase in size as they migrate to higher crypt positions.

C  Lgr5[+] colon SCs generate uniform clones of either enterocytes (left) or goblet cells (right). Tissues are from Lgr5-CreERT2-Confetti mice, 10 days post-induction.

D  The majority of clones emanating from Lgr5[+] cells are of uniform fate. Black bars show measurements of *n* = 119 clone induced in eight mice in the colon (Table EV2). Error bars are bootstrap standard deviations; gray bars and error bars show the theoretical prediction under late commitment (Materials and Methods).

E  Lineage tracing of arbitrary crypt cells using CAG-Cre/Confetti mice 3 days post-induction yields uniform clones consisting entirely of enterocytes (left) or goblet cells (right).

F  95% of the clones in CAG-Cre/Confetti mice are uniform in their fate (*n* = 71 clones). Error bars are bootstrap standard deviations.

G  Extensive overlap of proliferation and differentiation in the TA compartment. The white arrowheads mark Edu[+] and Ki67[+] cells in S phase, and yellow arrowhead points at a Ki67[+], Edu[−] cycling Gob5 cell.

Data information: Scale bars in (B, C, E, and G) are 10 μm. White dashed lines mark crypt borders (B, C, E, and G).

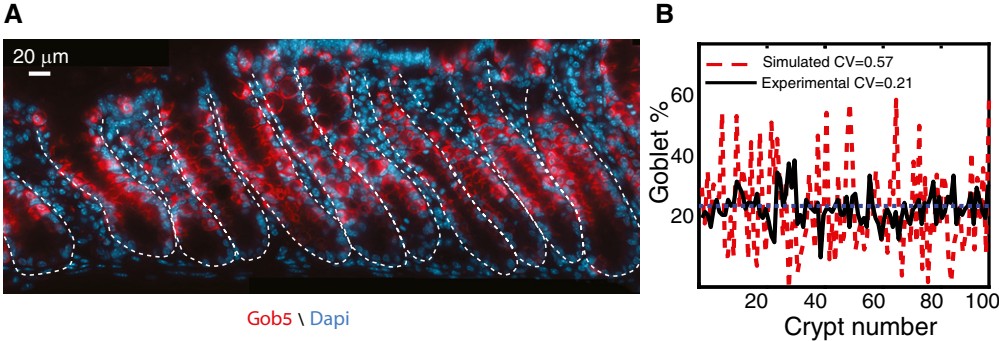

**Figure 3.  Variability in goblet cell proportions among different colonic crypts is lower than expected under stochastic early commitment.**

A   Representative image of the distal colon demonstrating the low variability in goblet cell proportions among crypts. Image composed of five stitched adjacent positions. White dashed lines mark crypt borders.

B   Variability in goblet cell proportions among colonic crypts is much smaller than expected based on stochastic early commitment. Red, goblet cell proportions in simulated crypts with stochastic early commitment. Black, measured proportions of goblet cells in 100 distal colon crypts of eight different mice. Dashed blue line marks the average proportion of goblet cells in both scenarios (0.25).

are responsive to Delta-Notch signaling from their neighbors. To this end, we sought to characterize the progenies of Lgr5$^+$ cells that operate in an environment where the potential for lateral inhibition is reduced (Fig 4C).

We crossed Lgr5-CreERT2 mice with Rosa26LSL-NICD mice that also harbor an inducible nuclear GFP label (Murtaugh *et al*, 2003). Tamoxifen induction yielded crypts with mosaic hyperactivation of Notch signaling and subsequent inhibition of Delta expression. The GFP nuclear label enabled tracing the progenies of induced Notch-hyperactive SCs as well as non-induced WT SCs (Fig 4C and D). Under stochastic cell autonomous commitment with no lateral inhibition, the proportions of secretory progenies of WT SCs should remain 25% (Fig 4D, left). In contrast, under lateral inhibition, non-induced SCs should produce more goblet cells in the mosaic crypts, where neighboring cells do not express Delta that would laterally inhibit them (Fig 4C and D right). Indeed, we found that non-induced SCs in the mosaic crypts yielded significantly more goblet cells compared to SCs in WT colonic crypts (59% versus 25%, $P < 10^{-13}$, Fig 4E and F). The less WT SCs per crypt, the higher was their probability to yield goblet cells (Spearman $R = 0.8$, $P < 10^{-23}$). Notably, while the average proportion of goblet cells among different crypts remained 25%, inter-crypt variability increased from 0.21 to 0.39 (Appendix Fig S5). These results demonstrate that lateral inhibition does operate in the colonic crypt and that it is important in minimizing fluctuations in cell proportions.

**Lateral inhibition is confined to the commitment zone**

Lateral inhibition operating throughout the crypt axis would give rise to mixed clones containing enterocytes (Notch) and goblet (Delta) cells, as Delta cells would inhibit their adjacent sibling cells after division (Fig 4B). Since mixed clones were rare (Fig 2C–F), we concluded that lateral inhibition must operate in the confined commitment zone. Indeed, we observed a strong depletion of pairs of adjacent Math1$^+$ cells at the immediate 2 crypt rows above the uppermost Lgr5$^+$ cell (Fig 5A and B, *P*-val < 10$^{-6}$). This pattern of depletion of neighboring Math1$^+$ cells was tightly restricted, as the number of Math1$^+$ cells with a neighboring Math1$^+$ cell dramatically increased already at 3–4

rows above the uppermost stem cell (> 60% in rows 3–4 versus < 20% in rows 1–2, Fig 5B).

To further test whether the 1:3 ratio of goblet cells and enterocyte is set up by lateral inhibition in the commitment zone, we measured the proportion of Gob5$^+$ cells in the two rows defined as the commitment zone and found them to be 25.6 ± 6.7% similar to the proportion in the differentiated compartment (25.5 ± 5.3, Fig 5C, rank-sum *P*-val = 0.85, Table EV4). Moreover, we examined the numbers of Gob5$^+$ uniform clones and Gob5$^-$ uniform clones and found a ratio of 1:3.2 (26 Gob5$^+$ clones versus 83 Gob5$^-$ clones, Table EV2). Thus, the 1:3 ratio of goblet cells and enterocytes seems to be determined by spatially restricted complete lateral inhibition operating at the two rows directly above the upper most Lgr5$^+$ cell. To assess the impact of lateral inhibition in the CZ on the reduction in variability in cell fate, we incorporated it into our crypt simulations. We observed a profound reduction in variability (CV = 0.26 Fig 5D and E), approaching the one observed in the colon. Thus, lateral inhibition operating in the commitment zone reduces the stochastic variability in cell proportions in the colon.

We next asked whether lateral inhibition in the CZ can also confer robustness in cell proportions to fluctuations in SC numbers. To obtain an estimate of the variability in the number of SCs per crypt, we counted the numbers of GFP$^+$ cells in the Lgr5-CreERT2 mice, which also harbors GFP as part of the knock-in cassette. We found that the number of GFP$^+$ cells per crypt is quite variable (17.5 ± 4.1, Table EV5), potentially introducing additional inter-crypt variability in cell production. The number of GFP$^+$ cells exceeds the estimated number of crypt SCs estimated from mathematical modeling of neutral drift processes (Kozar *et al*, 2013), since it is also expressed in the first TA progenitors (Kim *et al*, 2016). We incorporated variability in SC numbers into our simulations and found that a layer of lateral inhibition in the CZ renders the crypt almost completely insensitive to this variation (Appendix Fig S6A). We also relaxed other assumptions of our models and simulated crypts with different cell cycle distributions (Appendix Fig S6B and C), stem cell division rates (Appendix Fig S6D and E), and ratios of cell cycle periods between goblet cells and enterocytes (Appendix Fig S6F). We found that the reduction of noise in cell

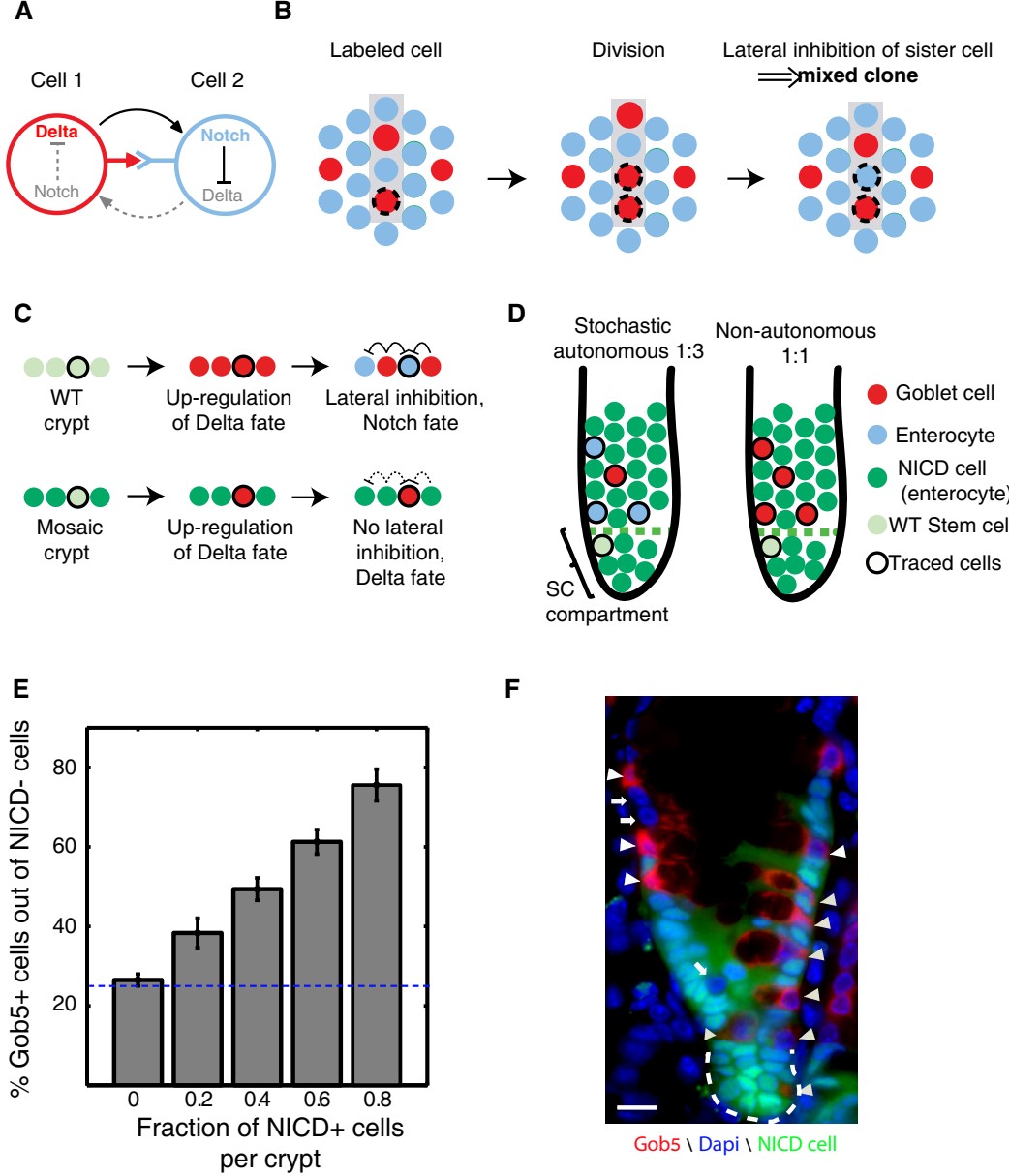

**Figure 4.  Cell fate in colonic crypts is determined non-autonomously by Delta-Notch-mediated lateral inhibition.**

A    A simplified scheme of Delta-Notch lateral inhibition. Upon binding of Delta ligand on cell 1 to Notch receptor on the adjacent cell 2, Notch pathway is activated in cell 2. This activation inhibits Delta ligand on cell 2 and thus deprives cell 1 from its Notch-activating signal, thus reinforcing its Delta expression state. Solid black arrows, active regulation; dashed gray arrows, inactive regulation.

B    Lateral inhibition operating throughout the crypt axis would generate mixed clones. Dashed circles mark the progenies of a labeled Delta cell (red). Upon division, the two Delta progenies would undergo lateral inhibition resulting in a trans-differentiation of one of the cells to a Notch fate (blue). This would lead to mixed clones, which are extremely rare in colonic crypts. Gray rectangle represents a column where division and subsequent cell movement have occurred.

C–F  WT cells in mosaic Lgr5-Cre/Rosa26LSL-NICD crypts compensate for Notch hyperactivation in their neighbors, generating secretory cells with increased probability and demonstrating that Delta-Notch lateral inhibition operates in colonic crypts. (C) Illustration of non-autonomous commitment through lateral inhibition. Upper row denotes the commitment events in a WT crypt when lateral inhibition is at play. Stem cell progenies undertake the default Delta fate (red), but subsequently laterally inhibit their neighbors through Notch signaling, yielding Notch-high enterocytes (blue). Bottom row demonstrates the expected outcome when neighbors of a traced WT stem cell are Notch hyperactive (dark green) and thus cannot deliver lateral inhibition (dashed inhibitory arrows). This would result in the traced cells preserving their Delta fate. (D) Expected fate outcomes in mosaic Notch-hyperactivated crypts without (left) and with (right) lateral inhibition. Under stochastic autonomous commitment (left), WT SCs generate goblet cells and enterocytes with a 1:3 ratio regardless of neighboring cells; however, under non-autonomous commitment (right), they generate predominantly goblet cells. (E) The proportion of goblet cells originating from WT SCs increases with the fraction of Notch-hyperactivated cells in the colonic crypt ($n = 100$ crypts, two mice, Spearman $R = 0.8$, $P < 10^{-23}$). Data were binned to groups according to the fraction of induced NICD cells in the crypt. Bars present the mean fraction of goblet cells out of the non-NICD-induced cells. Error bars are standard errors. Dashed horizontal blue line marks the 25% expected goblet cell proportion if the commitment were cell autonomous. (F) A mosaic crypt demonstrating non-autonomous commitment decision. Green, NICD cells. Red, goblet cells highlighted by arrowheads. Arrows point at WT enterocytes. Unlike the unperturbed tissue, where 25% of cells are goblet cells, 11/14 non-induced WT SCs in this crypt are goblet cells. White dashed lines mark crypt borders. Scale bar indicates 10 µm.

                                      

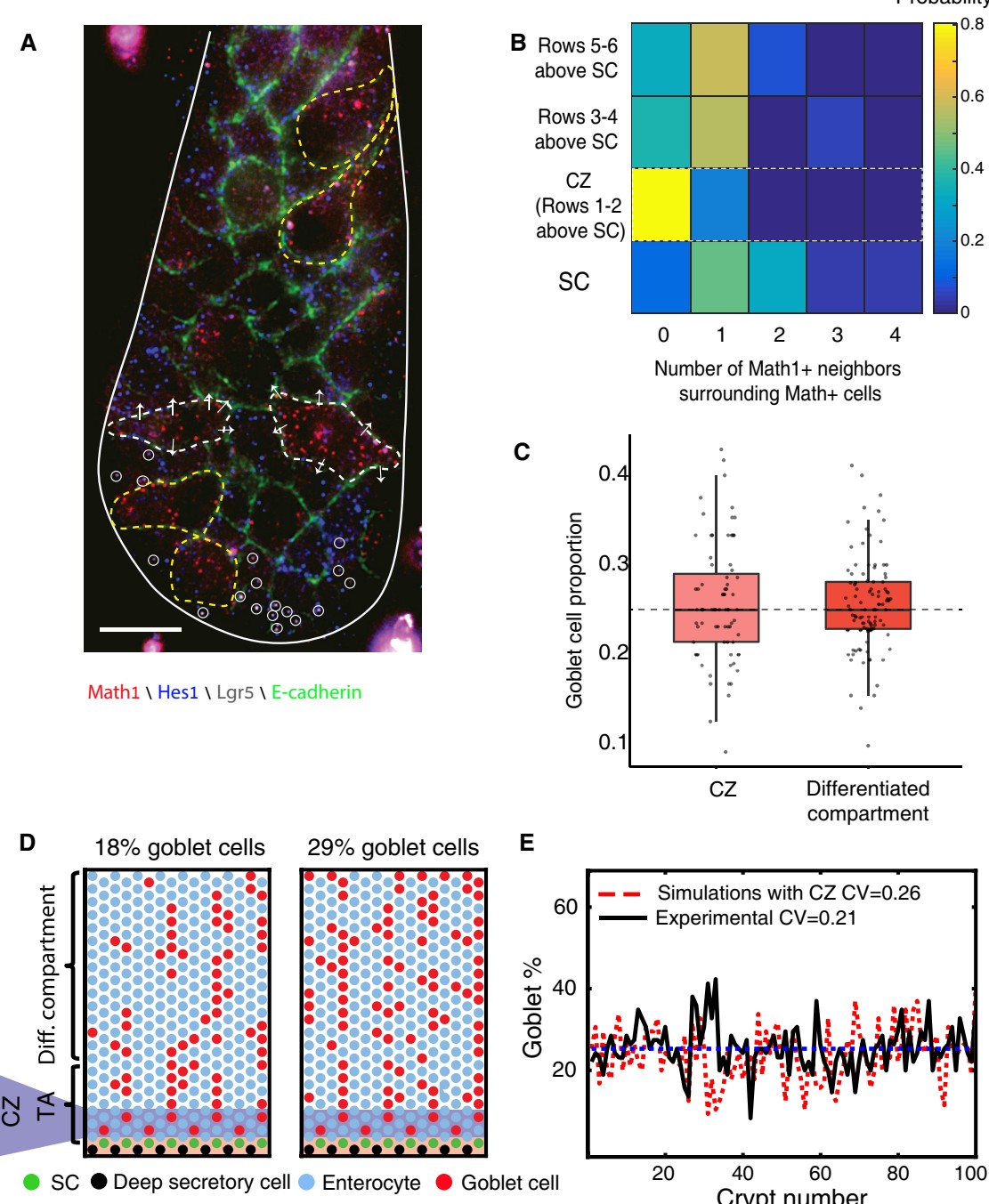

Math1 \ Hes1 \ Lgr5 \ E-cadherin

● SC  ● Deep secretory cell  ● Enterocyte  ● Goblet cell

**Figure 5.  Lateral inhibition in the commitment zone reduces variability in cell proportions in the colon.**

A   Math1[+] cells rarely have Math[+] neighbors in the commitment zone, in contrast to the SC compartment below and the TA compartment above. Red and blue dots are mRNA of Math1 and Hes1, respectively, visualized by smFISH. Dots marked with white circles are Lgr5 mRNA. Green is E-cadherin protein detected with simultaneous immunofluorescence. Math1[+] cells in CZ are marked with white dashed line. Arrows point at the Hes1-expressing neighbors. Neighboring Math1[+] cells in SC and TA compartments are marked with yellow dashed line. CZ was identified as the immediate two rows above the uppermost Lgr5-expressing cell. Scale bar, 10 μm.

B   Quantification of the fraction of neighboring Math1[+] cells at the stem cell compartment (22 cells), commitment zone (41 cells), 3–4 rows above the uppermost stem cell (16 cells), and 5–6 rows above the uppermost stem cell (12 cells). SCs were identified by simultaneous smFISH for Lgr5. N = 8 mice.

C   The 1:3 ratio of goblet cells and enterocytes observed in the commitment zone (light red, n = 75) is preserved in the differentiated compartment (dark red, n = 100). Note that the smaller sampling size in the commitment zone (due to the smaller number of rows) introduces greater sampling noise than in the differentiated compartment. Dashed line marks 25%. Data based on seven mice. Horizontal lines are medians, boxes demarcate the 25–75 percentiles, vertical lines are 1.5 times the inter-quartile range.

D   Two simulated crypts in which lateral inhibition operates in the commitment zone (CZ) spanning 2 rows above the SC zone.

E   Lateral inhibition in the CZ reduces variability in goblet cell proportions. Shown are the experimental data for the colon (black), simulated lateral inhibition in the CZ (red), and throughout the crypt axis (dashed blue).

proportions is insensitive to these changes. With lateral inhibition, 1:3 cell proportion is an emerging property of the hexagonal lattice geometry and the interactions between the cells within the two critical layers where lateral inhibition operates.

### Dispersive goblet cell migration minimizes noise accumulated in the TA compartment

Lateral inhibition at the commitment zone can yield a precise 3:1 enterocyte: goblet cell ratio; however, subsequent proliferation and migration in the TA compartment are stochastic events that can introduce yet additional variability. This is indeed evident in our simulations—the calculated CV after incorporating the commitment zone (~0.26) was somewhat higher than the CV we measured experimentally, suggesting that further mechanisms might be at play in addition to the commitment zone to ensure robust differentiation in the colonic crypt. The TA compartment above the commitment zone introduces additional noise to the goblet cell proportions, since divisions and migrations are random processes. Extensively high or low division rates of particular clones exiting the commitment zone can particularly bias these proportions in any given epithelial patch.

Goblet cell dispersal, through which clusters of adjacent goblet cells are broken up actively by absorptive cells, may alleviate this effect. If goblet cells are not immediately extruded from the crypt, but rather migrate more slowly and in a dispersive manner, their numbers would effectively average several stochastic rounds of TA cell production, reducing variability that may have accumulated during TA clonal expansion. Dispersive migration of goblet cells can also uniformly spread goblet cells by breaking patches (Bjerknes & Cheng, 1999; Togashi *et al*, 2011).

To assess the migration of goblet cells, we analyzed the height of goblet cell clones 14 days after Lgr5-Cre induction and found that it is significantly lower compared to enterocyte clones (11 ± 4 versus 24 ± 3, *P* = 0.003, Fig 6A, Table EV2). This result demonstrates that goblet cells migrate at a slower rate along the crypt vertical axis. To further establish the dispersive migration of goblet cells, we analyzed the abundance of neighboring goblet cell pairs along the crypt axis. The number of adjacent pairs increased in the TA compartment. This was expected, since we have shown that goblet cells divide without laterally inhibiting their neighbors in the TA compartment above the commitment zone. In contrast, the number of adjacent pairs gradually decreased in the differentiated compartment as expected from dispersive migration (Fig 6B, Appendix Fig S7, Appendix).

Slower dispersive migration of goblet cells should yield an increase in the ratio of goblet cells and enterocytes above the CZ. In contrast, our measurements indicated that the 1:3 ratio is preserved in the differentiated compartment (Figs 3 and 5C). Thus, the slower migration must be compensated by decreased division of goblet cells. Indeed, we found that goblet cells undergo less TA divisions than enterocytes, yielding smaller terminal clones (3 ± 1 cells per clone for goblet cell clones compared to 5 ± 2.5 cells per clone for enterocyte clones), thus preserving the 0.25 proportion of goblet cells established in the commitment zone (Fig 6C, Welch *t*-test *P*-val = 0.0025 Table EV2). The smaller terminal sizes of goblet cell clones could stem either from earlier cell cycle exit at lower crypt positions, or alternatively from slower cell cycle kinetics of goblet cells. To further characterize crypt cell cycle dynamics, we pulse labeled mice with EdU and analyzed the distribution of vertical crypt positions of EdU$^+$ goblet cells and EdU$^+$ enterocytes. We found no statistically significant difference between the crypt positions of these cycling cells (Appendix Fig S8, Welch test *P*-val = 0.28). Thus, goblet cells seem to compensate for their slower migration rates by cycling more slowly throughout the TA compartment. These results correlate with previous studies that demonstrated a smaller labeling index of goblet cells compared to the colon (Chang, 1971). Notably, our results highlight a difference between the colon and the small intestine, in which commitment to a secretory fate entails immediate exit from the cell cycle, rarely yielding clones with more than a single goblet cell (Stamataki *et al*, 2011). Fig 6D summarizes the proliferative features that maintain a tight 1:3 proportion among goblet: enterocyte cells in the colonic crypt.

An additional factor that can affect subsequent variability in cell proportions accumulated in the TA compartment is synchrony in division rates among clonal progenies (Hawkins *et al*, 2009). Synchronous divisions, where clonal progenies inherit their progenitor division rate, can yield extensively large or small clones, thus broadening the distribution of cell proportions (Fig 6E). We found through simulation that dispersive migration dramatically reduced variations emanating from synchronized divisions, also conferring robustness to such proliferative feature (Fig 6E and F, Appendix). Refining our computational model to include both lateral inhibition in the commitment zone and dispersive goblet cell migration captured the reduced inter-crypt variability of goblet cell proportions that we observed in our experiments (Fig 7A and B).

## Discussion

Our work highlighted a fundamental trade-off between timely production and robust differentiation of tissue stem cells. Timely production is facilitated by early commitment. It may be particularly important in the colonic crypts, since it serves to generate goblet cells at lower crypt positions, which may require mucosal protection from the hostile crypt microenvironment. In addition, early commitment facilitates ample time for cell maturation, a feature that becomes important when considering the relatively short lifetime of crypt cells above the TA compartment.

With early commitment, however, decisions are made by a small pool of progenitor cells, exposing the system to noise associated with small numbers. Delta-Notch lateral inhibition can generate robust proportions in static epithelial lattices, but is incompatible with early commitment in a dynamic epithelial sheet such as that of the colonic epithelium, where cells continuously acquire new neighbors while migrating upwards. Lateral inhibition operating throughout the crypt axis would render the differentiation process highly inefficient, as cells would continuously switch fates and trans-differentiate. Since many of the neighboring cells are siblings (Fig 2B, C and E, Appendix Fig S4), lateral inhibition would have yielded mixed clones, which we rarely observed (Fig 2D and F). This led us to identify the restriction of lateral inhibition to the commitment zone, where lateral inhibition fixes the 25% goblet cell proportion before TA divisions have initiated. In our simulations, we found that lateral inhibition operating in the commitment zone was of key importance in reducing the variability in goblet cell proportions among crypts (Fig 7A).

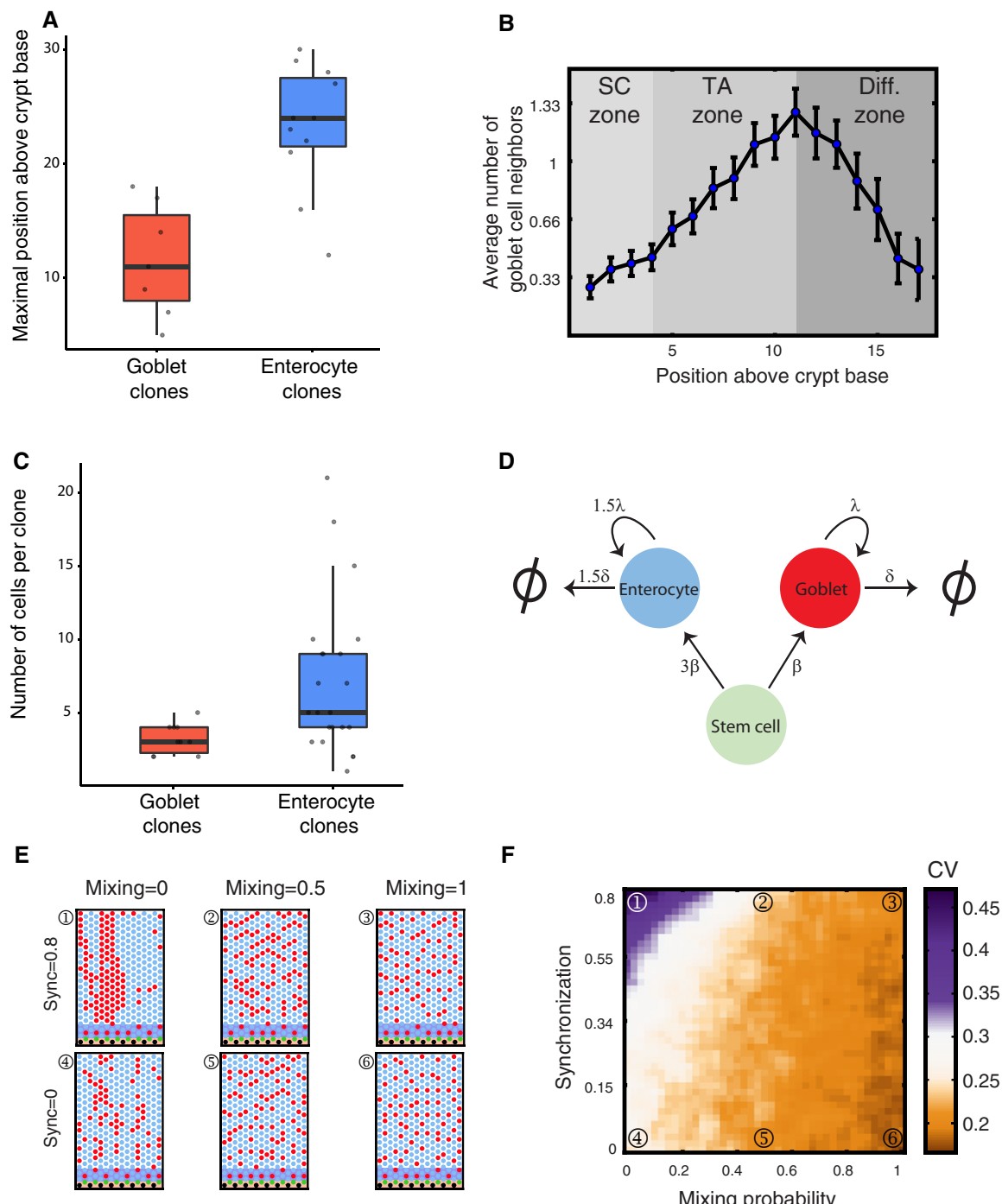

**Figure 6. Dispersive migration of goblet cells reduces variability in cell proportions in the colon.**

A   Goblet cells migrate substantially slower than enterocytes. Positions measured 14 days after tamoxifen induction (seven goblet cell clones and 11 enterocyte clones from two mice). Kruskal–Wallis test $P$ = 0.003. Horizontal lines are medians, boxes demarcate the 25–75 percentiles, vertical lines are 1.5 times the inter-quarentile range.

B   Goblet neighboring profile throughout the colon crypt vertical axis reveals a decrease in neighboring goblet cells above the TA zone, a hallmark of dispersive migration. Data were smoothed with a window of 5 crypt rows. Error bars are standard errors.

C   Goblet cells undergo less TA divisions. Bars represent 10 goblet cell clones and 22 enterocyte clones, from five mice induced by tamoxifen for 7, 10, and 14 days. Clones containing only post-mitotic cells were included (min 10 rows above crypt base). Welch *t*-test $P$ = 0.0025. Horizontal lines are medians, boxes demarcate the 25–75 percentiles, vertical lines are 1.5 times the inter-quarentile range.

D   A diagram summarizing the colonic crypt dynamics of cell production. Stem cells produce goblet cells and enterocytes in a ratio set by the commitment zone to be 1:3. λ, TA division rates; δ, extrusion rate from crypt. The slower extrusion of goblet cells is compensated by slower TA division rate.

E   Simulated crypts with high (top row) and low (bottom row) clonal synchronization and a range of dispersive migration rates (termed "mixing"). Numbered panels correspond to numbered regions in (F).

F   CV of goblet cell proportions among crypts decreases with dispersive migration and increases with synchronous divisions (see Appendix).

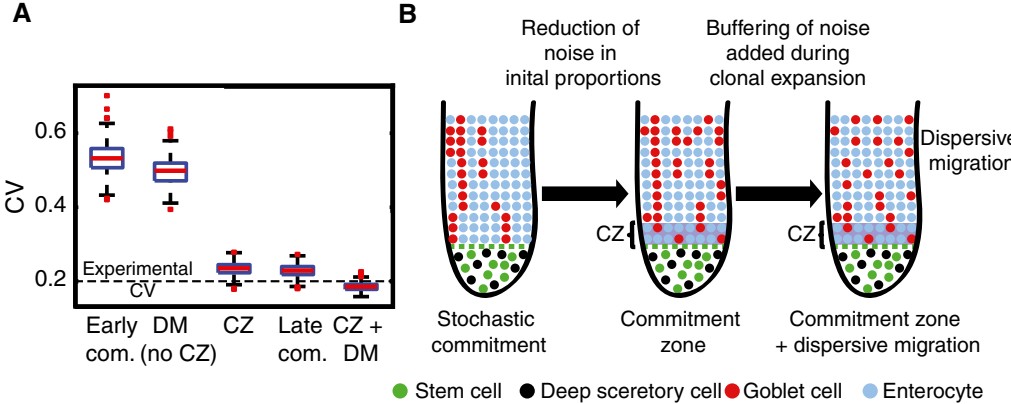

**Figure 7. Homeostatic mechanisms ensuring robust proportions in spite of early commitment.**

A  Summary graph showing the coefficient of variation (CV) of goblet cell proportions in different commitment strategies. DM, dispersive migration. CZ, spatially-restricted lateral inhibition in the commitment zone. Horizontal lines are medians, boxes demarcate the 25–75 percentiles, vertical lines are 1.5 times the inter-quarentile range.

B  Schematic illustration showing the mechanisms operating in the crypt to ensure a robust proportion of goblet cells.

The pattern of Delta-Notch lateral inhibition observed in our study is distinct from Delta-Notch-mediated feedback mechanisms operating in other stem cell-maintained tissues. Such mechanisms include Delta-Notch lateral inhibition operating between stem cells and their direct progenies, as in the fly gut (Ohlstein & Spradling, 2007), the mouse airway epithelium (Pardo-Saganta et al, 2015), and the mouse epidermis (Blanpain et al, 2006). The pattern also differs from the lateral inhibition between the small intestinal Paneth cells and their neighboring Lgr5$^+$ SCs. In contrast, we found that the feedback mechanism in the colonic crypt consists of lateral inhibition between SC progenies, confined to the commitment zone (Fig 5A and B). It would be important to identify the mechanisms facilitating the spatial restriction of Delta-Notch signaling. These could include modulators of Delta-Notch signaling (Noah & Shroyer, 2013), such as Musashi-1 (Kayahara et al, 2003), Numb (Yang et al, 2011), and Mir34a (Bu et al, 2013, 34), or cross talk between Notch and Wnt (Crosnier et al, 2006; Fre et al, 2009). An alternative explanation for the restricted lateral inhibition observed in the commitment zone could involve the timescales needed for its efficient implementation (Collier et al, 1996; Barad et al, 2010). Lateral inhibition may be more efficient in the lower TA layers, since cells spend more time there due to the low mitotic pressure they are exposed to (Fig 1D). In contrast, at higher crypt positions the larger mitotic pressure gives rise to cell acceleration, and as a result, cells constantly change their neighbors. The "mixed signals" they consequently sense in this more rapidly changing environment may not be consistent enough to facilitate fate switching.

The ratio between the secretory and enterocyte lineages could be modulated not only by the initial probability of SCs to yield progenies of either lineage, but also by the subsequent number of TA divisions and the rate of extrusion from the tissue. The 25% goblet cell proportion set up in the commitment zone could have been maintained if both lineages divided and migrated at similar rates. In contrast, we found that goblet cells are extruded more slowly from the crypts compared to enterocytes. This slower extrusion facilitates temporal averaging of noise in goblet cell production added during the stochastic divisions in the TA compartment. This effect is analogous to the reduction in cell-to-cell gene expression variability, stemming from transcriptional bursts, that is obtained when mRNA lifetimes are long (Raj et al, 2006; Eldar & Elowitz, 2010; Bahar Halpern et al, 2015). To compensate for the increase in goblet cell numbers, due to their longer dwell time in the crypt, we found that goblet cells divide less often, yielding smaller terminal clones, compared to enterocyte clones (Fig 6C and D).

In our simulations, we compared the two noise-reducing mechanisms—lateral inhibition in the commitment zone and slower dispersive migrations and found that the impact of dispersive migration was secondary to the initial noise reduction obtained through lateral inhibition, facilitating a noise reduction of 6% if it were to act alone, compared to 56% for lateral inhibition alone (Fig 7A). It will be important to explore the molecular features that enable slower goblet cell migration. These could be related to surface adhesion molecules that confer stronger contacts with the basal membrane for goblet cells, thus resisting mitotic pressure. Other options could be surface molecules that implement homotypic repulsive interactions with other goblet cells or heterotypic interactions with enterocytes. Similar heterotypic adhesion through the expression of nectins has been previously demonstrated in the inner ear (Togashi et al, 2011).

When analyzing the trade-off between early commitment and robust differentiation, one could have considered alternative tissue architectures in which post-mitotic cells undergo terminal differentiation before being made functional. In the colon, this would entail an additional compartment of cells where cell maturation would take place after late commitment. Such additional compartment would not change the flux of fully differentiated cells. Indeed, stratified epithelia such as the skin exhibit such design (Blanpain & Fuchs, 2006), where cells undergo terminal differentiation after exiting cell cycle. The colon differs from such examples in that the entire epithelium is exposed to the toxic microbiota-rich lumen, thus requiring mucosal protection. Given the limited diffusion of mucus (Atuma et al, 2001), early commitment and rapid maturation may be the only way to supply the required mucosal protection to the lower crypt regions.

Our findings of early commitment and robust differentiation may be relevant to other clonally expanding mammalian systems. Early commitment has been suggested to operate in the small intestinal epithelium (Cheng, 1974; Cheng & Leblond, 1974; Bjerknes & Cheng, 1981a, 1999); however, the considerably lower abundance of bacteria in the upper gastrointestinal tract may render it less sensitive to non-uniform mucosal coverage. Indeed, goblet cells are considerably more rare in the small intestine, comprising only 8% of the epithelium. Early commitment has also been suggested to operate in the hematopoietic system (Lu *et al*, 2011; Naik *et al*, 2013); however, its impact on variability in cell proportions may be dampened by the large numbers of hematopoietic stem cells making fate decisions (Sun *et al*, 2014; Busch *et al*, 2015). The trade-off between commitment stage and robustness could be particularly relevant to systems with smaller sizes, such as determination of lineages in early embryos (Morris *et al*, 2010; Yamanaka *et al*, 2010) and in small compartments during postnatal development (Itzkovitz *et al*, 2012). Our study provides a framework to examine trade-offs and optimality in these and in other mammalian systems.

# Materials and Methods

### Animal models

All animal studies were approved by the Institutional Animal Care and Use Committee of WIS. All mice were purchased from The Jackson Laboratory. To trace the fate of individual Lgr5$^+$ stem cells, Lgr5-EGFP-Ires-CreERT2 mice were crossed with R26R-Confetti mice (Snippert *et al*, 2010). Cre enzyme was induced in 8- to 12-week-old male mice by single intraperitoneal injections of 3 mg tamoxifen per 18 g mouse and mice were subsequently sacrificed for lineage tracing after 40 h, 2, 3, 4, 5, 7, and 14 days. For short-term lineage tracing, 8- to 12-week-old male mice carrying one CAG-Cre allele (Hayashi & McMahon, 2002; Lei & Spradling, 2013) and one R26R-Confetti allele were injected with 0.3 mg tamoxifen per 18 g mouse and subsequently sacrificed 3d after induction. Mosaic hyperactivation of Notch intracellular domain (NICD) in stem cells and in their progenies was achieved by crossing Lgr5-EGFP-Ires-CreERT2 and Rosa26LSL-NICD (Murtaugh *et al*, 2003). Double heterozygous 8-week-old males were sequentially injected three times with 3 mg tamoxifen per 18 g mouse every other day. Mice were sacrificed 14 days after the first injection. For EdU experiments, two C57BL/6 8-week-old male mice were injected with 300 μl 3 mM EdU and were sacrificed 1.5 h later.

### Tissue processing smFISH and imaging

Harvested colon tissues were flushed with cold 4% formaldehyde (FA) in PBS and incubated first in 4% FA/PBS for 3 h, then in 30% sucrose in 4% FA/PBS overnight at 4°C with constant agitation. Fixed tissues were embedded in OCT. All experiments were performed on the distal colon. In order to quantify full clone sizes (Table EV1), we used a Zeiss Axio Observer.Z1, equipped with a Spinning disk Yokogawa CSU-X1 and a Rolera EmC2 camera to image entire crypts in cryosections of 40–60 μm. For smFISH experiments, we used a Nikon-Ti-E inverted fluorescence microscope equipped with a Photometrics Pixis 1024 CCD camera to image 10–40 μm cryosection. Probe library design, hybridization procedures, and imaging settings were carried out according to published methods (Itzkovitz *et al*, 2011; Lyubimova *et al*, 2013). Goblet cells, enterocytes, and stem cells were identified using smFISH of the genes Gob5, Fabp2, and Lgr5, respectively. smFISH for Ki67 was used to identify proliferating cells (Table EV6). FITC-conjugated antibody for E-cadherin was added to the hybridization mix and used to visualize cell borders. EdU detection was performed using Click-iT EdU Alexa Fluor 488 Imaging Kit. Following the smFISH probes hybridization, tissue samples were washed twice with PBS for 5mins. Detection reaction was done according to the manufacturer's published protocol.

### Clonal analysis

To ensure that clones represented progenies of single Lgr5$^+$ cells, we analyzed the fraction of crypts that had labeling in either of the Confetti colors traced in the Lgr5-Cre/Confetti mice and found it to be rare (4.3 ± 3.1%). When calculating this fraction, we only considered crypts with GFP fluorescence, since the Lgr5-EGFP-Ires-CreERT2 knock-in cassette is patchy and is expressed in < 50% of the crypts in the distal colon. Out of the clones that had recombination, 45% were RFP, 25% were CFP, 28% were YFP, and 2% were GFP. When analyzing uniformity of fate in clones, we only considered clones that did not have any Lgr5$^+$ cell in them. These will consist, with high probability, of clones originating from Lgr5$^+$ cells that have been extruded from the stem cell compartment before having divided into it. To this end, we only analyzed the RFP and CFP clones, as YFP and GFP cannot be identified using the epifluorescence microscopy that we used for smFISH analysis, due to the Lgr5-GFP cassette.

### Statistical analysis

Kruskal–Wallis tests were performed on data unless otherwise stated. When calculating the expected number of mixed clones under late commitment (Fig 2D), we produced $10^6$ randomized datasets that included the same amount and sizes of the experimental clones (119 clones that had more than a single cell, Table EV2) from 8 mice at 5, 7, 10, and 14 days after tamoxifen induction. For each cell in the randomized dataset, we randomly assigned one of the two cell types (enterocytes and goblet cells), according to their frequency in the experimental dataset (0.76 and 0.24, respectively). We calculated the mean and standard deviation of the fraction of uniform clones in the randomized data. We used a *Z*-test on the fraction of uniform clones observed in the experiment to obtain *P*-value. Error bars in Fig 2D and F are bootstrap sampling of the data.

To compare the fraction of goblet cells within non-induced WT cells in NICD mosaic crypts with the corresponding fraction in WT crypts (Fig 4E), we performed a non-parametric one-sample Wilcoxon signed rank test against the null hypothesis of 0.25. For this analysis, 100 crypts were analyzed from two different NICD induced mice.

To quantify the number of Math1$^+$ neighboring cells of Math1$^+$ cells (Fig 5B), we only included Math1$^+$ cells for which all neighboring cells could be detected and stratified the results as follows:

SC for cells in the bottom of the crypt and up to the uppermost Lgr5$^+$ cells (identified by smFISH, $n = 22$), CZ for cells positioned in the two rows directly above the uppermost Lgr5$^+$ ($n = 41$), the third and fourth rows above the Lgr5$^+$ uppermost cell ($n = 16$), and the fifth and sixth rows above the Lgr5$^+$ uppermost cell ($n = 12$). Cells analyzed were from 8 different mice. Kruskal–Wallis tests were used to obtain the probability that the numbers of Math1$^+$ neighbors in the three compartments analyzed in Fig 5B (SC, CZ, 3–4 and 5–6) were drawn from the same distribution.

To compare between the sizes of goblet and enterocyte clones, and to compare between position of EdU-positive goblet cells and enterocytes, Welch *t*-tests were performed, to account for the unequal variances of the two groups.

### Crypt simulations

We modeled the crypt as a hexagonal lattice of cells covering a cylinder with 22 rows and 16 columns. Compartments included a single row of 8 SCs intermingled with 8 immotile deep secretory cells, six rows of TA proliferating cells, and 15 rows of differentiated cells. Cell division within the SC compartment could either be horizontal, resulting in displacement of one of the two adjacent SCs, or upwards in one of three directions. Division in the TA compartment resulted in progenies replacing one of the three higher cells. Each division gave rise to an upward shift of the entire column above the position where the progeny was placed and extrusion of the uppermost cell in the displaced column. Commitment to either goblet or enterocyte fates was stochastically set in the lower TA row (for early commitment, Fig 1C), lower differentiated compartment row (for late commitment, Fig 1B), or according to the commitment zone lateral inhibition pattern (Fig 5D). A detailed description of the model and simulations is provided in the Appendix.

**Expanded View** for this article is available online.

### Acknowledgements

We thank David Sprinzak, Uri Alon, Liran Shlush, and all members of our laboratory for valuable discussions. We thank Jakub Abramson, Elazar Zelzer, and Efi Massasa for experimental help with the mouse models used in this study. S.I. is supported by the Henry Chanoch Krenter Institute for Biomedical Imaging and Genomics, the Leir Charitable Foundations, Richard Jakubskind Laboratory of Systems Biology, Cymerman-Jakubskind Prize, the Lord Sieff of Brimpton Memorial Fund, the I-CORE program of the Planning and Budgeting Committee and the Israel Science Foundation (grants 1902/12 and 1796/12), the Israel Science Foundation grant No. 1486/16, the EMBO Young Investigator Program, and the European Research Council under the European Union's Seventh Framework Programme (FP7/2007-2013)/ERC grant agreement number 335122. S.I. is the incumbent of the Philip Harris and Gerald Ronson Career Development Chair.

### Author contributions

SI conceived the project. BT and SB-M performed all experiments and data analysis. AG performed the computational simulations with the supervision of SI and NB. All authors wrote the paper.

### Conflict of interest

The authors declare that they have no conflict of interest.

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
