## [Review Process File · Molecular Systems Biology]

Early commitment and robust differentiation in colonic crypts

Beata Toth, Shani Ben-Moshe, Avishai Gavish, Naama Barkai, Shalev Itzkovitz

Corresponding author: Shalev Itzkovitz, Weizmann Institute of Science

Review timeline:

Submission date:	25 August 2016
Editorial Decision:	29 September 2016
Revision received:	15 November 2016
Editorial Decision:	30 November 2016
Revision received:	30 November 2016
Accepted:	01 December 2016

Editor: Maria Polychronidou

Transaction Report:

1st Editorial Decision

29 September 2016

Thank you again for submitting your work to Molecular Systems Biology. We have now heard back from the two referees who agreed to evaluate your study. As you will see below, the reviewers appreciate that the presented findings seem interesting. However, they raise a number of concerns, which we would ask you to address in a revision of the study.

During our pre-decision cross-commenting process we have circulated the reports to both referees. In reply, reviewer #2 mentioned that most of the issues raised by reviewer #1 seem addressable. However, s/he sent the following comments regarding the recommendation of reviewer #1 to study the small intestine: "From available literature (scarce though) it is known that specification along absorptive versus secretory lineages works different in the small intestine. Therefore, the small intestine could not serve for verification of their model. The authors describe this in the text and I think they would be able to reason in a form of Discussion about the differences in different tissues." Along those lines, we think that it is not necessary to perform experiments to address this point.

Please feel free to contact me in case you would like to discuss further any of the referees' comments.

REFeree REPORTS

Reviewer #1:

Lgr5+ colon stem cells reside at the entire base of colonic crypts. The stem cells are actively cycling and generate progenitor cells on a daily basis. The progenitors undergo another few rounds of divisions while committing to either the enterocyte lineage or the secretory lineage of which goblet

cells are the most predominant member. Intriguing, stem cell self-renewal and cell fate decisions are mainly the result of stochastic processes, such as neutral competition for niche occupancy that determines stem cell self-renewal.

Considering the theoretical impact of stochasticity on noisy fluctuations of cellular proportions, Toth et al. investigate the surprisingly little variation in cellular composition between different crypts. The authors describe how timely production of differentiated cells via early commitment can go hand in hand with robust differentiation and low variability. By using both modeling and experimental data, they show that early commitment by itself results in high intercrypt variability but that this is reduced by lateral inhibition via Delta-Notch signaling in the commitment zone as well as dispersive goblet cell migration.

Major comments:

1- While the authors mainly discuss the existence or nonexistence of variability in cell proportions and mechanisms to reduce such fluctuations, they neglect the underlying mechanisms how ratios between cellular lineages are generated in the first place. How such ratios are established (e.g. the 1:3 ratio in colon versus 1:12 in small intestine) is an important fundamental question within developmental biology.

To understand the absence of variation on goblet cell numbers that exit the progenitor zone, it is essential to understand how many secretory vs absorptive cells are created the 'novo' and the extent of variation that is present at the onset of lineage specification. Therefore, the authors should quantify how many progenitor cells are generated by the Lgr5+ stem cell population per day and quantifying the fraction of these new cells (smFISH against Gob5?) that becomes restricted towards secretory vs enterocyte lineages. For instance, in the simulation the authors 'assigned' secretory or absorptive identity with a probability of 25 vs 75%. It might very well be that both lineages are in fact created with a 1:1 ratio (perfect LI in the circumference), followed by different number of cell cycles per lineage to generate different cell numbers per cell type (e.g. 1 to 2 cycles for secretory cells versus 3 to 4 for enterocytes). It is essential that the generation ratios and the average number of cell cycles per lineage (uniform clonal amplifications) are quantified using *in vivo* data. Moreover, the secretory-absorptive ratio in the small intestine is different than in the colon while overall crypt architecture is not dramatically different. Therefore, small intestine might serve as a very important verification of the model, i.e. do small intestinal stem cells generate less early secretory progenitors, or is it that the enterocyte progenitors in the small intestine undergo a couple of additional cell cycles in comparison to colon.

2- An important conclusion of the manuscript is the early commitment zone that is restricted to the very early progenitor cells (lower TA layers) that are positioned just above the Lgr5 stem cell zone. Within figure 5, it is unclear how the zones are exactly mapped. For instance, how is the stem cell zone defined? Moreover, since it is a central point in the manuscript, it is important to experimentally define the exact location of the CZ zone where LI is functional (for instance by using CAG-CreERT2/LSL-NICD mice to analyze the ability to transform cell fate of NICD induced cells along the crypt axes).

Minor comments:

- The model the authors use is a very simplistic representation of the *in vivo* crypt and it is based on multiple assumptions. For instance, the stem cell number is set at 8 with a division rate of 2.5 days. In fact, Lgr5+ stem cells divide each day, while their replacement rate is 2,5 days. Therefore, it is essential that the authors demonstrate how robust their modeling is and how minor changes of the variables (stem cell number, cell cycle speed, number of cell cycles per lineage, migration rate, etc) affects the outcome of the modeling.

- Differences in the number of cell cycles that lineages undergo might not be accompanied with slower cell cycle kinetics as is used in the simulations. Can the authors administer a 2hrs EdU pulse to quantify at which positions along the crypt axes goblet cell progenitors still undergo cell division? It might very well be that early goblet progenitors directly undergo one or two more cycles prior cell cycle exit. In that scenario, EdU+ goblet progenitors should only be detected at the lower half of the crypt.

Reviewer #2:

The manuscript by Toth et al. reports interesting and novel data showing mechanisms of colonic

stem cell differentiation during tissue homeostasis. For their study the authors use both mathematical modelling and mouse genetics tools. The study shows that fate commitment to either secretory or absorptive lineages occurs as soon as the colonic stem cell leaves its compartment. The authors further demonstrate that lateral inhibition operating via the Delta-Notch pathway is responsible for low variability in differentiated cell ratios in crypts. This mechanism is only functional in the "commitment zone" when the first cell fate choices are made. The different cell division rates between secretory and absorptive lineages, the speed of their extrusion and dispersive migration of goblet cells along the crypt further reduce variability of goblet-cell proportions within the crypt. This work is likely to be of interest to most geneticist and stem cell researchers. Therefore, I support it for publication.

Minor comment: It would be easier for readers of the manuscript to see the motivation for performing the experiments addressing dispersive migration of goblet cells if the authors introduce their explanations to the Results part from the Methods chapter. Page 51 "The crypt dynamics simulated above could not account for two features of goblet cells we observed - the decrease in the number of pairs of neighboring goblet cells in the differentiated compartment, where cells have ceased to divide (Fig 6C), and the slower goblet-cell migration rate compared to the absorptive cells (Fig 6A)."

1st Revision - authors' response

15 November 2016

Reviewers' comments:

Reviewer #1:

Lgr5+ colon stem cells reside at the entire base of colonic crypts. The stem cells are actively cycling and generate progenitor cells on a daily basis. The progenitors undergo another few rounds of divisions while committing to either the enterocyte lineage or the secretory lineage of which goblet cells are the most predominant member. Intriguing, stem cell self-renewal and cell fate decisions are mainly the result of stochastic processes, such as neutral competition for niche occupancy that determines stem cell self-renewal.

Considering the theoretical impact of stochasticity on noisy fluctuations of cellular proportions, Toth et al. investigate the surprisingly little variation in cellular composition between different crypts. The authors describe how timely production of differentiated cells via early commitment can go hand in hand with robust differentiation and low variability. By using both modeling and experimental data, they show that early commitment by itself results in high intercrypt variability but that this is reduced by lateral inhibition via Delta-Notch signaling in the commitment zone as well as dispersive goblet cell migration.

Major comments:

1- While the authors mainly discuss the existence or nonexistence of variability in cell proportions and mechanisms to reduce such fluctuations, they neglect the underlying mechanisms how ratios between cellular lineages are generated in the first place. How such ratios are established (e.g. the 1:3 ratio in colon versus 1:12 in small intestine) is an important fundamental question within developmental biology.

To understand the absence of variation on goblet cell numbers that exit the progenitor zone, it is essential to understand how many secretory vs absorptive cells are created the 'novo' and the extent of variation that is present at the onset of lineage specification. Therefore, the authors should quantify how many progenitor cells are generated by the Lgr5+ stem cell population per day and quantifying the fraction of these new cells (smFISH against Gob5?) that becomes restricted towards secretory vs enterocyte lineages. For instance, in the simulation the authors 'assigned' secretory or absorptive identity with a probability of 25 vs 75%. It might very well be that both lineages are in fact created with a 1:1 ratio (perfect LI in the circumference), followed by different number of cell cycles per lineage to generate different cell numbers per cell type (e.g. 1 to 2 cycles for secretory cells versus 3 to 4 for enterocytes). It is essential that the generation ratios and the average number of cell cycles per lineage (uniform clonal amplifications) are quantified using in vivo data.

We thank the reviewer for this important point. We have now quantified the fraction of goblet cells not only in the differentiated compartment at the upper crypt rows, but also in the

commitment zone, defined as the two rows directly above the stem cell compartment. We find that the fraction of goblet cells in the commitment zone is $25.6 \pm 6.5\%$, which is similar to the distribution we find at the differentiated zone (Kruskal Wallis p-val = 0.85). This new analysis is shown in Figure 5C and Table EV4.

Figure 5C. The 1:3 ratio of goblet cells and enterocytes observed in the commitment zone (light red, $n=75$) is preserved in the differentiated compartment (dark red, $n=100$). Note that the smaller sampling size in the commitment zone (due to the smaller number of rows) introduces greater sampling noise than in the differentiated compartment. Dashed line marks 25%. Data based on 7 mice.

In addition, as suggested by the reviewer, we now present the fraction of generated clones that are either Gob5+ or Gob5-. This data is based on Table EV2. This analysis demonstrates that the stem cells generate goblet cell clones and enterocyte clones at a ratio very close to 1:3. These new analyses are described on page 11, last paragraph:

“To further test whether the 1:3 ratio of goblet cells and enterocyte is set up by LI in the commitment zone we measured the proportion of Gob5+ cells in the 2 rows defined as the commitment zone and found them to be $25.6 \pm 6.7\%$ similar to the proportion in the differentiated compartment (25.5 ± 5.3 , Fig 5C, ranksum p-val=0.85, Table EV4). Moreover, we examined the numbers of Gob5+ uniform clones and Gob5- uniform clones and found a ratio of 1:3.2 (26 Gob5+ clones vs. 83 Gob5- clones, Table EV2). Thus the 1:3 ratio of goblet cells and enterocytes seems to be determined by spatially restricted complete LI operating at the 2 rows directly above the uppermost Lgr5+ cell.”

The additional differences we find in the number of cell cycles compensate for the differential migration rates. We describe the different mechanisms for shaping cell proportions in the new Figure Panel 6D, which summarizes panels 6A-C:

Figure 6D. A diagram summarizing the colonic crypt dynamics of the different cell types. Stem cells produce goblet cells and enterocytes in a ratio set by the commitment zone to be 1:3. λ - TA division rates, δ - extrusion rate from crypt. The slower extrusion of goblet cells is compensated by slower TA division rate.

This is discussed in the text on page 14, first paragraph:

“Slower dispersive migration of goblet cells should yield an increase in the ratio of goblet cells and enterocytes above the CZ. In contrast, our measurements indicated that the 1:3 ratio is preserved in the differentiated compartment (Fig 3, Fig 5C). Thus the slower migration must be compensated by decreased division of goblet cells. Indeed, we found that goblet cells undergo less TA divisions than enterocytes, yielding smaller terminal clones (3 ± 1 cells per clone for goblet-cell clones compared to 5 ± 2.5 cells per clone for enterocyte clones), thus preserving the 0.25 proportion of goblet cells established in the commitment zone (Fig 6C, Welch t-test p -val = 0.0025 Table EV2). The smaller terminal sizes of goblet cell clones could either stem from earlier cell cycle exit at lower crypt positions, or alternatively from slower cell cycle kinetics of goblet cells. To further characterize crypt cell cycle dynamics we pulse labeled mice with EdU and analyzed the distribution of vertical crypt positions of EdU+ goblet cells and EdU+ enterocytes. We found no statistically significant difference between the crypt positions of these cycling cells (Appendix Figure S8, Welch test = 0.28). Thus goblet cells seem to compensate for their slower migration rates by cycling more slowly throughout the TA compartment. These results correlate with previous studies that demonstrated a smaller labeling index of goblet cells compared to the colon (Chang, 1971). Notably, our results highlight a difference between the colon and the small intestine, in which commitment to a secretory fate entails immediate exit from the cell cycle, rarely yielding clones with more than a single goblet cell (Stamataki et al., 2011). Fig 6D summarizes the proliferative features that maintain a tight 1:3 proportion among goblet:enterocyte cell in the colonic crypt.”

Moreover, the secretory-absorptive ratio in the small intestine is different than in the colon while overall crypt architecture is not dramatically different. Therefore, small intestine might serve as a very important verification of the model, i.e. do small intestinal stem cells generate less early secretory progenitors, or is it that the enterocyte progenitors in the small intestine undergo a couple of additional cell cycles in comparison to colon.

We thank the reviewer for pointing this out. While goblet cell dynamics are different in the small intestine we now discuss previous work that quantified these features, demonstrating that goblet cells in the small intestine, unlike what we find in the colon, exit cell cycle immediately after specification. This is discussed on page 14, first paragraph:

“The smaller terminal sizes of goblet cell clones could either stem from earlier cell cycle exit at lower crypt positions, or alternatively from slower cell cycle kinetics of goblet cells. To further characterize crypt cell cycle dynamics we pulse labeled mice with EdU and analyzed the distribution of vertical crypt positions of EdU+ goblet cells and EdU+ enterocytes. We found no statistically significant difference between the crypt positions of these cycling cells (Appendix Figure S8, Welch test = 0.28). Thus goblet cells seem to compensate for their slower migration rates by cycling more slowly throughout the TA compartment. These results correlate with previous studies that demonstrated a smaller labeling index of goblet cells compared to the colon (Chang, 1971). Notably, our results highlight a difference between the colon and the small intestine, in which commitment to a secretory fate entails immediate exit from the cell cycle, rarely yielding clones with more than a single goblet cell (Stamataki et al., 2011).”

2- An important conclusion of the manuscript is the early commitment zone that is restricted to the very early progenitor cells (lower TA layers) that are positioned just above the Lgr5 stem cell zone. Within figure 5, it is unclear how the zones are exactly mapped. For instance, how is the stem cell zone defined?

We thank the reviewer for this comment; we clarify the definition of the CZ in the text on page 11:

“Since mixed clones were rare (Fig 2C-F) we concluded that LI must operate in the confined commitment zone. Indeed we observed a strong depletion of pairs of adjacent *Math1*+ cells at the immediate 2 crypt rows above the uppermost *Lgr5*+ cell (Fig 5A-B, $p\text{-val} < 10^{-6}$). This pattern of depletion of neighboring *Math1*+ cells was tightly restricted, as the number of *Math1*+ cells with a neighboring *Math1*+ cell dramatically increased already at 3-4 rows above the uppermost stem cell ($>60\%$ in rows 3-4 vs. $<20\%$ in rows 1-2, Fig 5B).“

This new analysis is shown in Fig 5B:

Figure 5B Quantification of the fraction of neighboring *Math1*+ cells at the stem cell compartment (22 cells), Commitment zone (41 cells), 3-4 rows above the uppermost stem cell (16 cells) and 5-6 rows above the uppermost stem cell (12 cells). SC were identified by simultaneous smFISH for *Lgr5*. $N=8$ mice.

In addition, we have modified Figure 5 panel A which now demonstrates the *Lgr5*+ cells in relation to the CZ:

Math1 \ Hes1 \ Lgr5 \ E-cadherin

Figure 5A *A Math1+ cells rarely have Math+ neighbors in the commitment zone, in contrast to the SC compartment below and the TA compartment above. Red and blue dots are mRNA of Math1 and Hes1, respectively, visualized by smFISH. Dots marked with white circles are Lgr5 mRNA. Green is E-cadherin protein detected with simultaneous immunofluorescence. Math1+ cells in CZ are marked with white dashed line. Arrows point at the Hes1 expressing neighbors. Neighboring Math1+ cells in SC and TA compartments are marked with yellow dashed line. CZ was identified as the immediate two rows above the uppermost Lgr5 expressing cell. Scale bar - 10mm.*

Moreover, since it is a central point in the manuscript, it is important to experimentally define the exact location of the CZ zone where LI is functional (for instance by using CAG-CreERT2/LSL-NICD mice to analyze the ability to transform cell fate of NICD induced cells along the crypt axes).

We thank the reviewer for this important point. We have now replaced Figure 5B with a panel that presents a more detailed analysis of the CZ according to the spatial adjacency patterns of the Math1+ cells (See above). This new data demonstrates that lateral inhibition is localized to the 2 rows directly above the uppermost Lgr5+ stem cell, as we find higher numbers of neighboring Math1+ cells already at the sequential two crypt rows (rows 3-4 above the uppermost SC).

This new analysis is discussed on page 11, second paragraph:

“This pattern of depletion of neighboring Math1+ cells was tightly restricted, as the number of Math1+ cells with a neighboring Math1+ cell dramatically increased already at 3-4 rows above the uppermost stem cell (>60% in rows 3-4 vs. <20% in rows 1-2, Fig 5B).”

We have now also generated CAG-CreERT2/LSL-NICD mice, as suggested by the reviewer, and attempted to assess the ability of cells to change fate from goblet cells to enterocytes upon NICD induction, with the aim of identifying differences in this propensity between the commitment zone and the upper crypt positions. We found only 2 GFP+Gob5+ cells out of 90 NICD+ cells throughout the crypt axis (Fig A1 below). Unfortunately this system does not have the power to reveal potential de-differentiation capacities for the following reasons: 1) Induction in the CAG-CreERT2/LSL-NICD mice requires several serial injections of tamoxifen. Thus analysis can only be made several days after the first induction, during which cells have already migrated from their original crypt position at induction. Given the dispersive goblet cell migration we observed in the colonic crypts it is very hard to determine the origin of the cells at induction by that time. 2) We found that a minimum of 2 days were required until GFP signal reached detectable levels. Since the lifetime of mRNA is on the order of a few hours we could thus miss goblet cells that have trans-differentiated preferentially in the commitment zone, since all of their mRNA would already have been degraded. While this mouse model did not have the statistical power to more tightly localize the commitment zone, we strongly feel that our new analysis in Fig 5B, performed at much more refined spatial resolution, clearly identifies the 2 immediate rows above the uppermost Lgr5+ cell as the commitment zone where LI occurs.

Figure A1 - CAG-cre/ Rosa26^{LSL-NICD} mice were injected 3 sequential days with 3mg tamoxifen and sacrificed at day 3. The fate of Notch hyperactive, NICD+ cells (green, marked with arrowheads) along the crypt axis was analyzed by Gob5+ smFISH (red). White dashed lines mark crypt borders. Scale bar - 5μm.

Minor comments:

- The model the authors use is a very simplistic representation of the in vivo crypt and it is based on multiple assumptions. For instance, the stem cell number is set at 8 with a division rate of 2.5 days. In fact, Lgr5+ stem cells divide each day, while their replacement rate is 2.5 days. Therefore, it is essential that the authors demonstrate how robust their modeling is and how minor changes of the variables (stem cell number, cell cycle speed, number of cell cycles per lineage, migration rate, etc) affects the outcome of the modeling.

We have now performed these additional simulations shown in Appendix Figure S6. We varied the SC numbers between 6-10, cell cycle period between 1 day and 5 days as well as the ratio of cell cycle periods between Delta and Notch cells. In addition we also tested the variability in cell proportions when cell cycles are not log-normally distributed but rather normally or exponentially distributed. We find that the reduction in variability in cell proportion is insensitive to these changes. These new simulations are shown in Appendix Figure S6:

Appendix Figure S6 - Lateral inhibition in the commitment zone confers robustness of goblet cell proportions to fluctuations of SC numbers and dynamics.

A Goblet cell proportions in 100 simulated crypts (dashed red line), each crypt contained a random number of stem cells, ranging from 6 to 10. Division times were randomly assigned from a lognormal distribution, with mean division time of $72\text{h} \pm 28.8\text{h}$. Division times for Notch and Delta cells were drawn from a lognormal distribution with mean division time of $47\text{h} \pm 15.66\text{h}$ and $72 \pm 28.8\text{h}$, respectively. Plotted in black in all panels **A-E** is the experimental measurement of goblet cell proportion in 100 crypts. Dashed blue line in panels **A-E** marks the expected goblet cell proportion under complete LI (25%).

B Goblet cell proportions in 100 simulated crypts (dashed red line), in which the division rate of the cells was drawn from a normal distribution. For SCs and goblet cells, mean division time was $72\text{h} \pm 11\text{h}$, for Notch cells, mean division time $47 \pm 7\text{hr}$.

C Goblet cell proportions in 100 simulated crypts (dashed red line), in which the division rate of the cells was drawn from an exponential distribution. Means of the exponential distributions are equal to those of the normal distributions.

D Goblet cell proportions in 100 simulated crypts (dashed red line), in which stem cells divide every 24 ± 9.6 h. Notch and Delta divisions were drawn as in **A**.

E Goblet cell proportions in 100 simulated crypts (dashed red line), in which stem cells divide once every 120 ± 61.53 h. Notch and Delta divisions were drawn as in **A**.

F Variability in goblet cell proportions is robust to different ratios in cell cycle periods of Delta and Notch cells. Notch cells divide every $47 \text{h} \pm 15.66$ h. Delta cells divide every 47 ± 15.66 , 70.5 ± 28.2 , 94 ± 42.72 , 117.5 ± 58.75 and 141 ± 74.21 h. SCs divide every 72 ± 28.8 h. For each division rate configuration, the CV of 100 simulations was calculated 50 times. Shown are the mean CV and the standard errors. Since slower Delta division rates were not compensated for by migration (as in main **Figure 6F**), they resulted in a lower mean and therefore a higher CV.

And discussed in the text on pages 12, second paragraph:

“We next asked whether LI in the CZ can also confer robustness in cell proportions to fluctuations in SC numbers. To obtain an estimate of the variability in the number of SCs per crypt we counted the numbers of GFP+ cells in the *Lgr5-CreERT2* mice, which also harbors GFP as part of the knock-in cassette. We found that the number of GFP+ cells per crypt is quite variable (17.5 ± 4.1 , Table EV5), potentially introducing additional inter-crypt variability in cell production. The number of GFP+ cells exceeds the estimated number of crypt SCs estimated from mathematical modeling of neutral drift processes (6-8, Kozar et al., 2013), since it is also expressed in the first TA progenitors (Kim et al., 2016). We incorporated variability in SC numbers into our simulations and found that a layer of LI in the CZ renders the crypt almost completely insensitive to this variation (Appendix Figure S6A). We also relaxed other assumptions of our models and simulated crypts with different cell cycle distributions (Appendix Figure S6B-C), stem cell division rates (Appendix Figure S6D-E) and ratios of cell cycle periods between goblet cells and enterocytes (Appendix Figure S6F). We found that the reduction of noise in cell proportions is insensitive to these changes. With LI 1:3 cell proportion is an emerging property of the hexagonal lattice geometry and the interactions between the cells within the two critical layers where LI operates.”

In addition, in Figure 6E-F we assessed the impact of migration rates (termed ‘Mixing’ on the noise in cell proportions). Indeed, we found that dispersive slower migration of goblet cells (high mixing) leads to a further reduction in the noise in goblet cell proportions compared to non-dispersive migration that is fueled by mitotic pressure, especially when divisions in crypts are synchronous within individual clones. This is discussed in the text on pages 12-15:

“Dispersive goblet cell migration minimizes noise accumulated in the TA compartment LI at the commitment zone can yield a precise 3:1 enterocyte:goblet cell ratio, however subsequent proliferation and migration in the TA compartment are stochastic events that can introduce yet additional variability. This is indeed evident in our simulations - the calculated CV after incorporating the commitment zone (~ 0.26) was somewhat higher than the CV we measured experimentally, suggesting that further mechanisms might be at play in addition to the commitment zone to ensure robust differentiation in the intestinal crypt. The TA compartment above the commitment zone introduces additional noise to the goblet cell proportions, since divisions and migrations are random processes. Extensively high or low division rates of particular clones exiting the commitment zone can particularly bias these proportions in any given epithelial patch. Goblet-cell dispersal, through which clusters of adjacent goblet cells are broken-up actively by absorptive cells, may alleviate this effect. If goblet cells are not immediately extruded from the crypt, but rather migrate more slowly and in a dispersive manner, their numbers would effectively average several stochastic rounds of TA cell production, reducing variability that may have accumulated during TA clonal expansion. Dispersive migration of goblet cells can also uniformly spread goblet cells by breaking patches (Bjerknes and Cheng, 1999; Togashi et al., 2011). To assess the migration of goblet cells we analyzed the height of goblet-cell clones 14 days after *Lgr5-Cre* induction and found that it is significantly lower compared to enterocyte clones (11 ± 4 vs. 24 ± 3 , $P=0.003$, Fig 6A, Table EV2). This result demonstrates that goblet cells migrate at a slower rate along the crypt vertical axis. To further establish the dispersive migration of goblet cells, we analyzed the abundance of neighboring goblet cell pairs along the crypt axis. The number of adjacent pairs increased in the TA compartment. This was expected, since we have shown that goblet cells divide without laterally inhibiting their neighbors in the TA compartment above the

commitment zone. In contrast, the number of adjacent pairs gradually decreased above the TA compartment as expected from dispersive migration (Fig 6B, Appendix Fig S7, Appendix). Slower dispersive migration of goblet cells should yield an increase in the ratio of goblet cells and enterocytes above the CZ. In contrast, our measurements indicated that the 1:3 ratio is preserved in the differentiated compartment (Fig 3, Fig 5C). Thus the slower migration must be compensated by decreased division of goblet cells. Indeed, we found that goblet cells undergo less TA divisions than enterocytes, yielding smaller terminal clones (3 ± 1 for goblet clones compared to 5 ± 2.5 for enterocyte clones), thus preserving the 0.25 proportion of goblet cells established in the commitment zone (Fig 6C, Welch t-test p-val = 0.0025 Table EV2). Smaller terminal size of goblet cells could either stem from earlier cell cycle exit at lower crypt positions for goblet cell clones, or alternatively from slower cell cycle kinetics. To address this we pulse labeled mice with EdU and analyzed the distribution of vertical crypt positions of EdU+ goblet cells and enterocytes were still in cycle and found no statistically significant difference (Appendix Figure S8, Welch test = 0.28). Thus goblet cells seem to compensate for their slower migration rates by cycling more slowly throughout the TA compartment. These results correlate with previous studies that demonstrated a smaller labeling index of goblet cells compared to the colon (Chang, 1971). Notably, our results highlight a difference between the colon and the small intestine, in which commitment to a secretory fate entails immediate exit from the cell cycle, rarely yielding clones with more than a single goblet cell (Stamatakis et al., 2011). Fig 6D summarizes the proliferative features that maintain a tight 1:3 proportion among goblet:enterocyte cell in the colonic crypt. An additional factor that can affect subsequent variability in cell proportions accumulated in the TA compartment is synchrony in division rates among clonal progenies (Hawkins et al., 2009). Synchronous divisions, where clonal progenies inherit their progenitor division rate, can yield extensively large or small clones, thus broadening the distribution of cell proportions (Fig 6E). We found through simulation that dispersive migration dramatically reduced variations emanating from synchronized divisions, also conferring robustness to such proliferative feature (Fig 6E-F, Appendix). Refining our computational model to include both LI in the commitment zone and dispersive goblet cell migration captured the reduced inter-crypt variability of goblet-cell proportions that we observed in our experiments (Fig 7A,B).”

- Differences in the number of cell cycles that lineages undergo might not be accompanied with slower cell cycle kinetics as is used in the simulations. Can the authors administer a 2hrs EdU pulse to quantify at which positions along the crypt axes goblet cell progenitors still undergo cell division? It might very well be that early goblet progenitors directly undergo one or two more cycles prior cell cycle exit. In that scenario, EdU+ goblet progenitors should only be detected at the lower half of the crypt.

We thank the reviewer for this important point. We have now administered EdU and sacrificed mice after 1.5 hours and performed simultaneous smFISH for KI67, Gob5 and Lgr5 in addition to EdU detection. We find that Goblet cells are in cycle throughout the TA zone rather than only in the CZ and that there is no differences between the positions of EdU+ goblet cells and EdU+ enterocytes. This new data is shown in Appendix Figure S8:

Appendix Figure S8 – Position of dividing goblet cells is indistinguishable from position of dividing enterocytes.

Replicating goblet cells are less frequent than replicating enterocytes (26 EdU-labeled Goblet cells and 219 EdU-labeled enterocytes), however the crypt positions of replicating goblet cells are not significantly different than positions of replicating enterocytes. $N=2$ mice. Mice were injected with EdU and sacrificed 1.5h later. EdU detection and smFISH for Gob5, Ki67 and Lgr5 were performed in parallel to identify the cycling cells, their position and their type.

and discussed in the text on page 14, first paragraph:

“The smaller terminal sizes of goblet cell clones could either stem from earlier cell cycle exit at lower crypt positions, or alternatively from slower cell cycle kinetics of goblet cells. To further characterize crypt cell cycle dynamics we pulse labeled mice with EdU and analyzed the distribution of vertical crypt positions of EdU+ goblet cells and EdU+ enterocytes. We found no statistically significant difference between the crypt positions of these cycling cells (Appendix Figure S8, Welch test = 0.28). Thus goblet cells seem to compensate for their slower migration rates by cycling more slowly throughout the TA compartment. These results correlate with previous studies that demonstrated a smaller labeling index of goblet cells compared to the colon (Chang, 1971).”

We have also replaced Figure 2G with a new panel showing both Ki67 and EdU staining, demonstrating dividing goblet cells in higher crypt positions:

G Extensive overlap of proliferation and differentiation in the TA compartment. The white arrowheads mark Edu+ and Ki67+ cells in S phase, yellow arrowhead points a Ki67+, Edu- cycling Gob5 cell.

Reviewer #2:

The manuscript by Toth et al. reports interesting and novel data showing mechanisms of colonic stem cell differentiation during tissue homeostasis. For their study the authors use both

mathematical modelling and mouse genetics tools. The study shows that fate commitment to either secretory or absorptive lineages occurs as soon as the colonic stem cell leaves its compartment. The authors further demonstrate that lateral inhibition operating via the Delta-Notch pathway is responsible for low variability in differentiated cell ratios in crypts. This mechanism is only functional in the "commitment zone" when the first cell fate choices are made. The different cell division rates between secretory and absorptive lineages, the speed of their extrusion and dispersive migration of goblet cells along the crypt further reduce variability of goblet-cell proportions within the crypt. This work is likely to be of interest to most geneticist and stem cell researchers. Therefore, I support it for publication.

Minor comment: It would be easier for readers of the manuscript to see the motivation for performing the experiments addressing dispersive migration of goblet cells if the authors introduce their explanations to the Results part from the Methods chapter. Page 51 "The crypt dynamics simulated above could not account for two features of goblet cells we observed - the decrease in the number of pairs of neighboring goblet cells in the differentiated compartment, where cells have ceased to divide (Fig 6C), and the slower goblet-cell migration rate compared to the absorptive cells (Fig 6A)."

We thank the reviewer for this suggestion. We have now incorporated this explanation in the Results section on page 13, first paragraph:

"LI at the commitment zone can yield a precise 3:1 enterocyte:goblet cell ratio, however subsequent proliferation and migration in the TA compartment are stochastic events that can introduce yet additional variability. This is indeed evident in our simulations - the calculated CV after incorporating the commitment zone (~0.26) was somewhat higher than the CV we measured experimentally, suggesting that further mechanisms might be at play in addition to the commitment zone to ensure robust differentiation in the intestinal crypt. The TA compartment above the commitment zone introduces additional noise to the goblet cell proportions, since divisions and migrations are random processes. Extensively high or low division rates of particular clones exiting the commitment zone can particularly bias these proportions in any given epithelial patch. Goblet-cell dispersal, through which clusters of adjacent goblet cells are broken-up actively by absorptive cells, may alleviate this effect. If goblet cells are not immediately extruded from the crypt, but rather migrate more slowly and in a dispersive manner, their numbers would effectively average several stochastic rounds of TA cell production, reducing variability that may have accumulated during TA clonal expansion. Dispersive migration of goblet cells can also uniformly spread goblet cells by breaking patches (Bjerknes and Cheng, 1999; Togashi et al., 2011)."

2nd Editorial Decision

30 November 2016

Thank you again for submitting your work to Molecular Systems Biology. We have now heard back from reviewer #1 who was asked to evaluate your revised study. As you will see below, this reviewer is satisfied with the modifications made and thinks that the study is now suitable for publication in Molecular Systems biology.

Before we formally accept your paper, we would ask you to address some remaining editorial issues listed below.

REFEREE REPORT

Reviewer #1:

The manuscript of Itzkovitz and colleagues significantly improved and contains a thorough analysis how theoretical noisy fluctuations in cell type abundance in the colon are limited in practice. The work is of wide interest to many stem cell and developmental biologist.

As a verification of their own model, it's unfortunate that the authors didn't perform measurements, quantifications and analysis of relative cell type abundance in the small intestine themselves rather than relying on other's work. Nevertheless, I strongly support publication of this manuscript in its current state.

Corresponding Author Name: Shalev Itzkovitz

Manuscript Number: : MSB-16-7283